**Estimating CCN number concentrations using aerosol optical properties:**

**Role of particle number size distribution and parameterization**

Yicheng Shen[1,2], Aki Virkkula [2,1,3], Aijun Ding[1], Krista Luoma[2], Helmi Keskinen [2,4],

Pasi P. Aalto [2],   Xuguang Chi[1], Ximeng Qi[1], Wei Nie[1], Xin Huang[1],

Tuukka Petäjä[2,1], Markku Kulmala[2], Veli-Matti Kerminen[2]

[1]Joint International Research Laboratory of Atmospheric Sciences, School of Atmospheric Sciences,

Nanjing University, Nanjing, 210023, China

[2]Institute for Atmospheric and Earth System Research, University of Helsinki, Helsinki, Finland

[3]Finnish Meteorological Institute, Helsinki, Finland

[4]Hyytiälä Forestry Field Station, Hyytiäläntie 124, Korkeakoski FI 35500, Finland

Keywords: CCN, scattering coefficient, backscatter fraction, Ångström exponent

**Abstract**

The concentration of cloud condensation nuclei (CCN) is an essential parameter affecting aerosol-cloud interactions within warm clouds. Long-term CCN number concentration ($N_{CCN}$) data are scarse, there are a lot more data on aerosol optical properties (AOPs). It is therefore valuable to derive parameterizations for estimating $N_{CCN}$ from AOP measurements. Such parameterizations have been made earlier, and in the present work a new one is presented. The relationships between $N_{CCN}$, AOPs and size distributions were investigated based on in-situ measurement data from six stations in very different environments around the world. The relationships were used for deriving a parameterization that depends on the scattering Ångström exponent (SAE), backscatter fraction (BSF) and total scattering coefficient ($\sigma_{sp}$) of PM10 particles. The analysis first showed that the dependence of $N_{CCN}$ on supersaturation SS can be described by a logarithmic fit in the range SS < 1.1%, without any theoretical reasoning. The relationship between $N_{CCN}$ and AOPs was parameterized as: $N_{CCN} \approx ((286 \pm 46)\text{SAE} \ln(\text{SS}/(0.093 \pm 0.006))(\text{BSF} - \text{BSF}_{min}) + (5.2 \pm 3.3))\sigma_{sp}$, where $\text{BSF}_{min}$ is the minimum BSF, in practice the 1st percentile of BSF data at a site to be analyzed. At the lowest supersaturations of

each site (SS ≈ 0.1%), the average bias, defined as the ratio of the AOP-derived and measured $N_{CCN}$, varied from ~0.7 to ~1.9 at most sites except at the Himalayan site PGH where the bias was > 4. At SS > 0.4% the average bias ranged from ~0.7 to ~1.3 at most sites. For the marine-aerosol dominated site ASI the bias was higher, ~1.4 – 1.9. In other words, at SS > 0.4% $N_{CCN}$ was estimated with an average uncertainty of approximately 30% by using nephelometer data. The biases were mainly due to the biases in the parameterization related to the scattering Ångström exponent SAE. The squared correlation coefficients between the AOP-derived and measured $N_{CCN}$ varied from ~0.5 to ~0.8. To study the physical explanation of the relationships between $N_{CCN}$ and AOPs, lognormal unimodal particle size distributions were generated and $N_{CCN}$ and AOPs were calculated. The simulation showed that the relationships of $N_{CCN}$ and AOPs are affected by the geometric mean diameter and width of the size distribution and the activation diameter. The relationships of $N_{CCN}$ and AOPs were similar to those of the observed ones.

**1. Introduction**

Aerosol-cloud interactions (ACI) are the most significant sources of uncertainty in estimating the radiative forcing of the Earth's climate system (e.g., Forster et al., 2007; Kerminen et al., 2012), which makes it more challenging to predict the future climate change (Schwartz et al., 2010). An essential parameter affecting ACI within warm clouds is the concentration of cloud condensation nuclei (CCN), i.e., the number concentration of particles capable of initiating cloud droplet formation at a given supersaturation. Determining CCN concentrations and their temporal and spatial variations is one of the critical aspects to reduce such uncertainty.

CCN number concentrations ($N_{CCN}$) have been measured at different locations worldwide (e.g., Twomey, 1959; Hudson,1993; Kulmala et al., 1993; Hämeri et al., 2001; Sihto et al., 2011; Pöhlker et al., 2016; Ma et al., 2014). However, the accessible data especially for long-term measurements are still limited in the past and nowadays due to the relatively high cost of instrumentation and the complexity of long-term operating. As an alternative to direct measurement, $N_{CCN}$ can also be estimated from particle number size distributions and chemical composition using the Köhler equation. Several studies have investigated the relative

importance of the chemical composition and particle number distributions for the estimation of $N_{CCN}$ (Dusek et al., 2006; Ervens et al., 2007; Hudson, 2007; Crosbie et al., 2015). For the best of our understanding, particle number size distributions are more important in determining $N_{CCN}$ than aerosol chemical composition. This makes particle number size distribution measurements capable of serving as a supplement to direct CCN measurements.

Considering the tremendous spatiotemporal heterogeneity of atmospheric aerosol, neither direct measurements of $N_{CCN}$ nor the concentrations estimated from particle size distributions are adequate for climate research. In order to overcome the limitation of current measurements, many studies have attempted to estimate $N_{CCN}$ using aerosol optical properties (AOPs) (e.g., Ghan et al., 2006; Shinozuka et al., 2009; Andreae, 2009; Jefferson, 2010; Liu et al., 2014; Shinozuka et al., 2015; Tao et al., 2018). This approach would give both geographically wider and temporally longer estimates of $N_{CCN}$ than the available particle number size distribution and direct CCN measurement data. For instance, on 20 June 2019 the WMO Global Atmosphere Watch World Data Centre for Aerosols (GAW-WDCA) (http://ebas.nilu.no/) contained particle number size distribution data sets from 22 countries altogether from 58 stations, but only 5 of them were outside Europe. The CCN counter (CCNC) data were from 3 European sites. On the other hand, in the same data base, the light scattering coefficients measured with a nephelometer were from 31 countries and 103 stations located on all continents and also on some islands. The temporal coverage data in the GAW-WDCA data base is such that the oldest nephelometer data, those from Mauna Loa, start in 1974, whereas the oldest particle number size distribution data, those from the SMEAR II station in Finland, start in 1993. Another easily available source for data is the US Department of Energy Atmospheric Radiation Measurement (ARM) user facility (https://www.arm.gov/data). On 20 June 2019 we found that the ARM research facility data contained particle size distribution data from 7 permanent sites and light scattering coefficient measured with a nephelometer from 20 sites. It is clear that there are other data sets of all of these measured around the world, but those that can be found either from the GAW-WDCA or the ARM data bases are quality controlled and readily available.

Most of the above-mentioned studies attempted to link $N_{CCN}$ with extensive AOPs, such as the

aerosol extinction coefficient ($\sigma_{ext}$), aerosol scattering coefficient ($\sigma_{sp}$) and aerosol optical depth
(AOD). Both $N_{CCN}$ and $\sigma_{sp}$ are extensive properties that vary with a varying aerosol loading.
The most straightforward approach to estimate CCN is to utilize the ratio between CCN and
one of the extensive AOPs (e.g. AOD, $\sigma_{ext}$, $\sigma_{sp}$). However, the ratio is not a constant. Previous
studies have also pointed out that the relationship between $N_{CCN}$ and extensive AOPs are
nonlinear. On one hand, Andreae (2009) reported that the relationship between AOD at the
wavelength $\lambda$ = 500 nm ($AOD_{500}$) and CCN number concentration at the supersaturation of 0.4%
($CCN_{0.4}$) can be written as $AOD_{500}=0.0027 \cdot (CCN_{0.4})^{0.640}$, which indicates that AOD and CCN
depend in a non-linear way on each other: for a larger AOD there are more CCN per unit change
in AOD. On the other hand, Shinozuka et al. (2015) indicated that the larger the extinction
coefficient $\sigma_{ext}$ was, the fewer CCN were per unit change of $\sigma_{ext}$.
Some studies have also involved intensive aerosol optical properties, such as the scattering
Ångström exponent (SAE), hemispheric backscattering fraction (BSF) and single-scattering
albedo (SSA) to build up a bridge between the $N_{CCN}$ and AOPs. Jefferson (2010) used BSF and
SSA to parameterize the coefficients $C$ and $k$ in the relation $N_{CCN}(SS) = C \times (SS)^k$, where SS is
the supersaturation percent (Twomey, 1959) and the exponent $k$ is a function of SSA which
means it depends both on the scattering and absorption coefficients. Liu and Li (2014) discussed
how different aerosol properties affect the ratio of $N_{CCN}$ to $\sigma_{sp}$, i.e., $R_{CCN}/\sigma_{sp}$ based on *in-situ*
and remote-sensing data. Shinozuka et al. (2015) used SAE and aerosol extinction coefficient
to estimate $N_{CCN}$. Tao et al. (2018) used a novel method to derive the ratio $R_{CCN}/\sigma_{sp}$ which they
named as $AR_{sp}$, based on SAE and aerosol hygroscopicity using a humidified nephelometer.
All the studies mentioned above noted that the particle number size distribution (PNSD) plays
an important role in estimating $\underline{N}_{CCN}$ from aerosol optical properties.
In this paper we will analyze the relationships between $N_{CCN}$, aerosol optical properties and size
distributions at six different types of sites around the world. The relationships obtained from
the field sites will be used for developing a parameterization for calculating $N_{CCN}$ using AOPs.
We will also study the physical explanations of the relationships between $N_{CCN}$ and AOPs by
simulations.

## 2. Methods

### 2.1 Sites and measurements

In-situ measurements of AOPs, particle number size distributions (PNSDs), and $N_{CCN}$ were conducted at SMEAR II in Finland, SORPES in China, and 4 ARM Climate Research Facility (ACRF) sites (Mather and Voyles, 2013). The locations and measurement periods are listed in Table 1.

The Station for Measuring Forest Ecosystem-Atmosphere Relations (**SMEAR II**) is located at the Hyytiälä Forestry Field Station (61°51' N, 24°17' E, 181 m above sea level) of University of Helsinki, 60 km north-east from the nearest city. The station represents boreal coniferous forest, which covers ~8 % of the Earth's surface. Total scattering coefficient ($\sigma_{sp}$) and hemispheric backscattering coefficient ($\sigma_{bsp}$) of sub-1 μm and sub-10 μm particles are measured using a TSI-3563 3-wavelength integrating nephelometer at $\lambda$ = 450, 550, and 700 nm. The calibration, data processing and calculation of AOPs followed the procedure described by Virkkula et al. (2011) and Luoma et al. (2019). $N_{CCN}$ was measured at the supersaturations (SS) of 0.1%, 0.2%, 0.3%, 0.5% and 1.0% using a DMT CCN-100 CCN counter, similar to Schmale et al. (2017). A whole measurement cycle takes around 2 hours; data were interpolated to hourly time resolution to compare with other measurements. PNSDs were measured with a custom-made Differential Mobility Particle Sizer (DMPS) system in size range 3–1000 nm (Aalto et al., 2001). A more detailed description of CCN measurements and station operation can be found in Sihto et al. (2011) and Paramonov et al. (2015).

The Station for Observing Regional Processes of the Earth System (**SORPES**) is located in a suburb of Nanjing, a megacity in the Yangtze River Delta municipal aggregation (32°07'14" N, 118°57'10" E; ~40m a.s.l.). $\sigma_{sp}$ and $\sigma_{bsp}$ of total suspended particles (TSP) were measured with an Ecotech Aurora-3000 3-wavelength integrating nephelometer at $\lambda$ = 450, 525, and 635 nm as described by Shen et al. (2018). $N_{CCN}$ was measured using a CCN-200 dual column CCN counter at 5 supersaturations: 0.1%, 0.2%, 0.4%, 0.6% and 0.8%. The two columns make the same cycle simultaneously to cross-check with each other. Each cycle took 30 minutes. PNSDs in the size range of 6 - 800 nm were measured with a DMPS built by University of Helsinki.

More details of the measurements at SORPES are given by, e.g., Ding et al. (2013, 2016) and Qi et al. (2015).

The US Atmospheric Radiation Measurement Mobile Facility (AMF) measures atmospheric aerosol and radiation properties all over the world. The first AMF (AMF1) was deployed in 2005 with both a CCN counter and a nephelometer. Between 2011 and 2018, AMF1 was operated at four locations: Ganges Valley (**PGH**) in the Himalayas, Cape Cod, Massachusetts (**PVC**) in a coastal area of U.S., Manacapuru (**MAO**) downwind of the city of Manaus, Brazil, and Ascension Island (**ASI**) on the South Atlantic Ocean downwind from Africa. Three of them were accompanied by a scanning mobility particle sizer (SMPS; Kuang, 2016). The SMPS is also part of the Aerosol Observing System (AOS) running side by side with AMF1 since 2012. Both PNSDs and AOPs are available simultaneously at PVC, MAO, and ASI. $\sigma_{sp}$ and $\sigma_{bsp}$ of sub-1 μm and sub-10 μm particles are measured at all AMF1 locations by integrating nephelometers (Uin, 2016a). The size range of the SMPS is around 11 – 465 nm with slightly different ranges for different periods. $N_{CCN}$ is measured at different supersaturations, with the details given in Table 1. The supersaturations are typically calibrated before and after each campaign at an altitude similar to measurement site according to the CCN handbook (Uin, 2016b). Detailed information about each dataset and measurement site can be found in the AOS handbook (Jefferson, 2011) or ARM web site (http://www.arm.gov/) and references thereby.

Ganges Valley (PGH) is located in one of the largest and most rapidly developing sections of the Indian subcontinent. The aerosols in this region have complex sources, including coal combustion; biomass burning; automobile emissions; and dust. In monsoon seasons, dust dominates the aerosol mass due to transportation (Dumka et al., 2017; Gogoi et al., 2015).

PVC refers to the on-shore data set for the 'first column' of the Two-Column Aerosol Project (TCAP) on Cape Cod, Massachusetts, USA. This is a coastal site but also significantly affected by anthropogenic emissions (Berg et al., 2016).

MAO refers to Manacapuru in Amazonas, Brazil. It is a relatively clean site where Manaus

pollution plumes and biomass burning plumes impact the background pristine rainforest aerosol
alternately (e.g., de Sá et al., 2019).
Ascension Island (ASI) is located in the southeast Atlantic where westward transport of
biomass-burning aerosols from southern Africa may increase aerosol concentrations to high
levels. Air mass at this site usually a mixture with aged biomass-burning plume and sea-salt
aerosol. The aerosol loading can be very low when there is no pollution plume. In this case,
there is a substantial uncertainty on the backscatter fraction.
The primary purpose of this study is to use as basic and readily accessible measurement data as
possible to estimate $N_{CCN}$. Aerosol optical properties are measured at different cutoff diameters,
usually 1 μm, 2.5 μm, 10 μm or TSP. At several stations there are two sets of AOPs using two
cutoff diameters. For this study we chose to use AOP data with the 10 μm cutoff (if data for
both 10 μm and 1 μm are available) that is more commonly used than smaller cutoff diameters.
**2.2 Data processing**
Regardless of the time resolution of raw data, all the data in this study were adjusted into hourly
averages before further analyses. Suspicious data within the whole dataset were removed
according to the following criteria:
1) for the size distributions, all the data with unexplainable spikes were removed manually;
2) for CCN measurements, insufficient water supply may cause underestimation of CCN,
especially at lower supersaturations (DMT, 2009). $N_{CCN}$ reading at lower SS has a sudden drop
a few hours before the similar sudden drop for higher SS under such conditions, so data from
such periods were removed;
3) if any obvious inconsistencies between the AOPs and PNSD or between the $N_{CCN}$ and PNSD
were found on closure study, all the data in the same hour were removed.
Special treatments were carried out for the ASI dataset. There will inevitably be a considerable
uncertainty in the backscattering fraction if the zero point of either $\sigma_{sp}$ or $\sigma_{bsp}$ is inaccurate in
very clean conditions. The measured $\sigma_{sp}$ was in agreement with that calculated from the PNSD
with the Mie model. However, in the data $\sigma_{bsp}$ approaches 0.3 Mm$^{-1}$ whenever $\sigma_{sp}$ approaches
0. Thus, we subtracted from back scattering coefficients a constant 0.3 Mm$^{-1}$ and no longer used
any data points with $\sigma_{sp} < 2$ Mm$^{-1}$ for this site to assure the data quality.
A more detailed description of the total number of available hourly-averaged data, accepted
data and removed data and the fractions of these are presented the supplement S1.
**2.3 Optical properties calculated from the nephelometer data**
The hemispheric backscatter fraction BSF was calculated from

$$\text{BSF} = \frac{\sigma_{bsp}}{\sigma_{sp}} \tag{1}$$

where $\sigma_{sp}$ and $\sigma_{bsp}$ are the total scattering coefficient and backscattering coefficient, respectively.
BSF depends on both particle size and shape. For very small particles, BSF approaches the
value of 0.5 and decreases with an increasing particle size (e.g., Wiscombe and Grams, 1976;
Horvath et al., 2016; Shen et al., 2018). Jefferson (2010) used BSF as a proxy for the particle
size for estimating CCN concentrations from in situ AOP measurements.
Scattering Ångström exponent (SAE) was calculated from total scattering coefficients $\sigma_{sp}$ at
wavelengths $\lambda_1$ and $\lambda_2$ from

$$\text{SAE} = -\frac{\log(\sigma_{sp}(\lambda_1)) - \log(\sigma_{sp}(\lambda_2))}{\log(\lambda_1) - \log(\lambda_2)} \tag{2}$$

For those sites where the TSI 3563 nephelometer was used the wavelength pair was 450 nm
and 700 nm, for the Ecotech Aurora-3000 nephelometer the wavelength pair was 450 nm and
635 nm. SAE is typically considered to be associated with the dominating particle size. Its large
values (e.g. SAE>2) indicate a large contribution of small particles, whereas small values (e.g.
SAE<1) indicate a large contribution of large particles. SAE can be retrieved by remoting
sensing measurements and it serves as a proxy for particle size for satellite (e.g., Higurashi and
Nakajima, 1999; King et al., 1999; Liu et al., 2008) and sunphotometry (e.g., Holben et al.,
2001; Gobbi et al., 2007) retrieval of aerosol optical properties, even though it is well known
that this is just a crude approximation. Many studies found that this relationship is not
unambiguous. Surface mean diameter (SMD) and volume mean diameter (VMD) correlate well
with SAE while geometric mean diameter (GMD) correlates poorly with SAE according to
Schuster et al. (2006), Virkkula et al. (2011) and Shen et al. (2018).
The reason for calculating both BSF and SAE in the present work is that they provide
information on the particle size distribution, yet being sensitive to slightly different particle size
ranges (e.g., Andrews et al., 2011; Collaud Coen et al., 2007). A detailed model analysis by
Collaud Coen et al., 2007) showed that BSF is more sensitive to small accumulation mode
particles, i.e., particles in the size range <400 nm whereas SAE is more sensitive to particles in
the size range of 500–800 nm.

### 2.4 Light scattering calculated from the particle number size distributions

Light scattering coefficients (both $\sigma_{sp}$ and $\sigma_{bsp}$) were calculated using the Mie code similar to
Bohren and Huffman (1983). The refractive index was set to the average value of 1.517+0.019i
reported for SMEAR II by Virkkula et al. (2011). The wavelength for Mie modeling was set to
550 nm, which is the same as in the measurements. The whole size range of the DMPS or the
SMPS, depending on the station, was used. BSF was calculated from (1) by using the modeled
$\sigma_{sp}$ and $\sigma_{bsp}$. Both the size range and the selected constant refractive index create uncertainty
especially when the modeled scattering is compared with scattering of PM10 aerosols. However,
the purpose of the modeled scattering was quality control and removal of inconsistent data.

### 2.5 CCN number concentration calculated from the particle number size distribution

The $\kappa$-Köhler theory uses a single parameter $\kappa$ to describe the relationship between
hygroscopicity and water vapor saturation (Petters and Kreidenweis, 2007).

$$S(D) = \frac{D^3 - D_d^3}{D^3 - D_d^3(1-\kappa)} exp\left(\frac{4\sigma_{s/a}M_w}{RT\rho_W D}\right) \tag{3}$$

Here $S$(D) is water vapor saturation, which equals to SS+100%, $D$ is the diameter of the wet
particle, $D_d$ is particle dry diameter and $\kappa$ is the hygroscopicity parameter. The rest of the
coefficients are usually set to constant, for instance in this study, $\sigma_{s/a} = 0.072$ J/m$^2$ is the surface
tension of the solution/air interface, $R= 8.314$ J/mol is the universal gas constant, $T=298$K is
temperature, $\rho_w=1000$kg/m$^3$ is the density of water, $M_w=0.018$kg/mol is the molecular weight
of water. At given $\kappa$ and $D_d$, $S(D)$ is a function of the wet diameter $D$, which is physically larger
than $D_d$. As a combination of the Kelvin effect and the Raoult effect, $S(D)$ first increases and
then decreases as $D$ increases, and there is a maximum value for $S(D)$ in the $S$-$D$ curve. Here,
we call the maximum value of $S(D)$ and corresponding $D$ as $S(D)_{max}$ and $D_{max}$ respectively.
Physically, if $S(D)_{max}$ larger than the SS of the environment, the dry particle will reach a wet
diameter $D$ between $D_d$ and $D_{max}$; while if $S(D)_{max}$ is smaller than the SS of the environment the
dry particle can grow to infinite sizes, which means it is a so-called activated particle. $S(D)_{max}$
decreases monotonically as $D_d$ increases. Thus we can iterate $D_d$ until $S(D)_{max}$ equals to a given
SS. We call this $D_d$ the critical diameter $D_m$. Particles with $D_d>D_m$ their $S(D)_{max} <$ SS and they
can be activated while the smaller particles cannot.
Under the assumption of fully internally mixed particles, the CCN number concentration
calculated from the particle number size distributions ($N_{CCN}$(PNSD)) is obtained by integrating
the PNSD of particles larger than the critical dry particle diameter ($D_m$):

$$N_{CCN}(PNSD) = \int_{D_m}^{\infty} n(\log D_p) d\log D_p \qquad (4)$$

at a given SS. All particles with a diameter larger than $D_m$ can act as CCN. We calculated
$N_{CCN}$(PNSD) at the supersaturations at which CCN were measured in the different stations (e.g.,
0.1%, 0.2%, 0.3%, 0.5% and 1.0% for SMEAR II ).
The accuracy of $N_{CCN}$(PNSD) is affected by the treatment of $\kappa$. In this study, we are not trying
to achieve an accurate value of $\kappa$ but instead want to illustrate that even an arbitrary setting of
$\kappa$ can yield reasonable CCN concentrations. This approach is named as 'unknown chemical
approach' in (Kammermann et al., 2010) and as 'Prediction of $N_{CCN}$ from the constant $\kappa$' in
Meng et al., (2014). Both of them give a detailed discussion of how this approach performs.
Arbitrary $\kappa$ is not performing as good as a proper $\kappa$ when calculating $N_{CCN}$ , yet we believe that
it is good enough to be an alternative to measuring CCN in the empirical estimation of this

study. Wang et al. (2010) also claimed that $N_{CCN}$(PNSD) may be successfully obtained by assuming an internal mixture and using bulk composition few hours after emissions. For SORPES, ASI and PVC, we simply set a global-average value of 0.27 for $\kappa$ (Pringle et al., 2010; Kerminen et al., 2012). For the forest sites, SMEAR II and MAO, we set $\kappa = 0.12$, which is close to the value of $\kappa$ for Aitken mode particles reported previously by studies at forest sites (Sihto et al., 2011; Hong et al., 2014). Here we used $N_{CCN}$(PNSD) for quality control and removal of inconsistent data.

**2.6 Aerosol optical properties and CCN concentrations of simulated size distributions**

For studying the relationships of particle size, $N_{CCN}$ and AOPs, we generated unimodal particle number size distributions n(GMD,GSD) with varying the geometric mean diameter (GMD) and geometric standard deviation (GSD). For them we calculated the same AOPs with the Mie model as were obtained from the real measurements from the stations $\sigma_{sp}$ and $\sigma_{bsp}$ and from these the BSF at the wavelengths $\lambda = 550$ nm. $N_{CCN}$ was calculated simply by integrating number concentrations of particles larger than a critical diameter of 50 nm, 80 nm, 90 nm, 100 nm, and 110 nm, and 150 nm. When the global average hygroscopicity parameter $\kappa = 0.27$ is used this corresponds to a SS range of ~0.14% − 0.74%.

Using a unimodal size distribution for the simulation is an approximation. In the boundary layer, particle number size distributions consist typically of an Aitken mode in the size range of ~25 − 100 nm, an accumulation mode in the size range of 100 − 500 nm and, following atmospheric new particle formation, also a nucleation mode in the size range of < 25 nm (e.g., Dal Maso et al., 2005; Herrmann et al., 2015; Qi et al., 2015). While the particle number concentration is dominated by the smaller modes, essentially all light scattering is due to the accumulation mode and also coarse particles in the range of 1 - 10 µm. For example, at SMEAR II the average contribution of particles smaller than 100 nm to total scattering was ~0.2 % and even at the end of new particle formation events it was no more than ~2% (Virkkula et al., 2011). Also most of the CCN are in the accumulation mode size range, especially at low supersaturations (SS < 0.2%); at higher SS also Aitken mode particles contribute to CCN (Sihto et al., 2011).

**3. Relationships between $N_{CCN}$ and AOPs**

We first present general observations of the $N_{CCN}$ and AOPs at all the six sites and investigate in more detail data from SMEAR II. Based on the relationships of AOPs and $N_{CCN}$ at SMEAR II, we further use data from all the stations and develop a simple and general combined parameterization for estimating $N_{CCN}$.

**3.1 Site-dependent $N_{CCN}$ - AOP relationships**

The averages of AOPs of $PM_{10}$ particles and $N_{CCN}$ at four supersaturations during the analyzed period for each site are presented in Table 2. In general all of them are cleaner than SORPES and more polluted than SMEAR II, based on the average values of $\sigma_{sp}$. The average values of $N_{CCN}$ are obviously higher in more polluted air as well as can be seen in the values presented in Table 2. The dependence of $N_{CCN}$ on SS is shown by plotting the averages of the measured $N_{CCN}$ at the six sites at the station-specific supersaturations of the CCN counters (Fig. 1). In all these different types of environments a logarithmic function fits better to the data than the power function $N_{CCN}(SS) = C \times (SS)^k$. It is not a new observation that the power function is not perfect for describing the $N_{CCN}$ vs. SS relationship. Also other function types have been used in the literature, for instance a product of the power function and the hypergeometric function (Cohard et al., 1998; Pinsky et al., 2012), an exponential function (Ji and Shaw, 1998; Mircea et al., 2005; Deng et al., 2013) and the error function (e.g., Dusek et al., 2003 and 2006b; Pöhlker et al., 2016). In the following analysis of the relationships between $N_{CCN}$, AOPs and SS we will use logarithmic fittings to the data without any theoretical reasoning.

Since there is obviously a positive correlation between the averages of $N_{CCN}$ and $\sigma_{sp}$ in Table 2, it is reasonable to study whether this is true also for the hourly-averaged data. A scatter plot shows that the correlation between $N_{CCN}$ and $\sigma_{sp}$ was weak at SMEAR II, especially for higher supersaturations (Fig 2). In spite of this, when the scatter plots are color-coded with respect to BSF, the relationship between $N_{CCN}$ and $\sigma_{sp}$ becomes clear: $N_{CCN}$ grows almost linearly as a function of $\sigma_{sp}$ for a narrow range of values of BSF. This indicates BSF can serve as a good proxy for describing the ratio between $N_{CCN}$ and $\sigma_{sp}$.

Hereafter, we will use the term $R_{CCN/\sigma} = N_{CCN}/\sigma_{sp}$ to describe the relationship between $N_{CCN}$ and
$\sigma_{sp}$, similar to Liu and Li (2014). Note that this same ratio was defined as $AR_{scat}$ in Tao et al.
(2018). $R_{CCN/\sigma}$ varies over a wide range of values, so a proper parameterization to describe it is
of significance.
The first step in the development of the parameterization was to calculate linear regressions of
$R_{CCN/\sigma}$ vs BSF. $R_{CCN/\sigma}$ depends clearly on BSF (Fig. 3) as

8                 $R_{CCN/\sigma} = a\,BSF + b$                                      (5)

The correlation between BSF and $R_{CCN/\sigma}$ is strong when $\sigma_{sp} > 10$ Mm$^{-1}$. At $\sigma_{sp} < 10$ Mm$^{-1}$ the
uncertainty of the nephelometer is higher, which may at least partly explain the lower
correlation. Based on this we used $\sigma_{sp} > 10$ Mm$^{-1}$ as the criterium for the data fitting.
Linear regressions of $R_{CCN/\sigma}$ vs BSF were applied to data from all the analyzed stations. For
each dataset and individual supersaturation, $a$ and $b$, i.e. the slope and offset of the linear
regression, haves different value as presented in Table 3. The calculation of $a$ and $b$ are based
on data with $\sigma_{sp} > 10$ Mm$^{-1}$ only. The following discussion is based on the ordinary linear
regression (OLR). In addition, we repeated the calculations with the Reduced Major Axis
(RMA) regression, see supplement S2.
The parameterization gives the formula for calculating $N_{CCN}(AOP)$, ie, $N_{CCN}$ calculated from
measurements of AOPs:

22               $N_{CCN}(AOP_1) = (a_{SS}BSF + b_{SS}) \cdot \sigma_{sp}$                          (6)

The subscript 1 for $AOP_1$ indicates the first set of parameterization.
Scatter plots of $N_{CCN}(AOP_1)$ vs $N_{CCN}(meas)$ are presented for two supersaturations, high and
low, at the six stations (Fig. 4). The correlation coefficient $R^2$ between $N_{CCN}(AOP_1)$ and
$N_{CCN}(meas)$ is higher at lower supersaturations than that at higher supersaturations in most of
the scatter plots shown in Fig. 4. A reasonable explanation for this is that the higher the
supersaturation is, the smaller are the particles that can act as CCN. And further, the smaller the
particles are, the less they contribute to both total scattering and backscattering and the higher
is the relative uncertainty of both of them and thus also the uncertainty of $N_{\mathrm{CCN}}(\mathrm{AOP}_1)$.
**3.2 Site-independent relationships between $N_{\mathrm{CCN}}$, AOPs and supersaturations**
The relationships between $N_{\mathrm{CCN}}$ and AOPs are obviously different for each site and
supersaturation. We next try to find a way to combine them into a site-independent form. First,
the slopes and offsets obtained from the linear regression (Table 3) were plotted as a function
of SS (Fig 5). The data obviously depend logarithmically on SS, so that (6) becomes
$$N_{CCN}(AOP_2) = \left( a_{SS}BSF + b_{ss} \right)\sigma_{sp} = \left( \left( a_1 \ln(SS) + a_0 \right)BSF + b_1 \ln(SS) + b_0 \right)\sigma_{sp} \qquad (7)$$
The coefficients $a_0$, $a_1$, $b_0$ and $b_1$ obtained from the regression of $a_{\mathrm{SS}} = a_1\ln(\mathrm{SS}) + a_0$ and $b_{\mathrm{SS}} =$
$b_1\ln(\mathrm{SS}) + b_0$ vs. the supersaturations SS for each station are presented in Table 4.
Note that also a power function of SS of the form $\mathrm{SS}^k$ was used for fitting the data (Fig 5). This
is the dependence on SS assumed for instance in the parameterization by Jefferson (2010). It is
obvious that the power function fitting is not as good as the logarithm of SS. This is in line with
the fittings to $N_{\mathrm{CCN}}$ vs. SS (Fig. 1) and the related discussion in section 3.1.
The relationships of the coefficients in Table 4 are next used to get a combined, more general
parameterization. Obviously the $a_0$ vs. $a_1$, $b_0$ vs. $b_1$, $a_1$ vs. $b_1$ and $b_0$ vs. $b_0$ pairs from all stations
follow very accurately the same lines (Fig 6). Linear regressions yielding $a_0 = (2.38 \pm 0.06)a_1$,
$b_0 = (2.33 \pm 0.03)b_1$, and $b_1 = (\text{-}0.096 \pm 0.013)a_1 + (6.0 \pm 5.9)$ were used, after the simple algebra
in the supplement S3, to get
$$N_{CCN}(AOP_2) \approx \left( \ln(SS) + (2.38 \pm 0.06) \right)\left( a_1(BSF - (0.096 \pm 0.013)) + (6.0 \pm 5.9) \right)\sigma_{sp}$$
$$\approx \ln\left( \frac{SS}{0.093 \pm 0.006} \right)\left( a_1(BSF - (0.096 \pm 0.013)) + (6.0 \pm 5.9) \right)\sigma_{sp} \qquad (8)$$

where both the coefficient $a_1$ and the constant $6.0 \pm 5.9$ have units of $[N_{\mathrm{CCN}}]/[\sigma_{sp}] = \mathrm{cm}^{-3}/\mathrm{Mm}^{-1}$.
This is the general formula for the parameterization. In both (7) and (8) the only unquantified
coefficient is now $a_1$. However, we can find some ways to quantify also it.
The above derivation of the combined parameterization by using the logarithms of SS was fairly
straightforward. In the error-function parameterizations of Dusek et al. (2003) and Pöhlker et
al. (2016) there are adjustable parameters that affect the argument of the error function. In the
parameterization of Ji and Shaw (1998) there is an exponential function where the argument
contains the power function of SS and the parameterization of by Cohard et al. (1998) is a
product of the power function and the hypergeometric function. If these functions were used
for fitting the $N_{CCN}$(AOP, SS) data it would be would be more complicated to combine the site-
dependent parameterizations into a general equation analogous to Eq. (8). The simplicity of the
logarithmic fitting makes it most suitable for our approach. The disadvantage of Eq. (8) is that
it predicts no upper limit for $N_{CCN}$ at high supersaturations. This is not correct since $N_{CCN}$ cannot
be larger than the total particle number concentration and therefore it has to be emphasized that
the parameterization presented here is only valid in the range of SS < 1.1%.
For a given station, if there are simultaneous data of $N_{CCN}$(meas) and $\sigma_{sp}$ for some reasonably
long period, (8) can be adjusted. To estimate what is a reasonably long period, we added an
analysis in the supplement S5. It shows that when the number of hourly samples is > ~1000,
the uncertainty in $BSF_{min}$ is low enough. Instead of subtracting (0.096 ± 0.013) from BSF, the
minimum BSF = $BSF_{min}$ in the data set will be used. Further, when BSF = $BSF_{min}$ the factor
$a_1(BSF - BSF_{min}) = 0$ and $N_{CCN}(AOP_2) \approx R_{min} \cdot \sigma_{sp}$ where $R_{min}$ is the minimum $R_{CCN/\sigma}$ in the data
set. It follows that
$$N_{CCN}(AOP_2) \approx \left( a_1 \ln\left( \frac{SS}{0.093 \pm 0.006} \right)(BSF - BSF_{min}) + R_{min} \right)\sigma_{sp} \qquad (9)$$
The derivation of (9) is shown in the supplement S4. In the data processing the 1st percentiles
of both BSF and $R_{CCN/\sigma}$ are used as $BSF_{min}$ and $R_{min}$, respectively. Here the free parameters are
$a_1$, $BSF_{min}$ and $R_{min}$.
The coefficient $a_1$ is positively correlated with SAE. The linear regressions of $a_1$ and the average
and median scattering Ångström exponent of $PM_{10}$ particles (SAE) (Table 4) at the 6 sites in
the analyzed periods yield $a_1 \approx (298 \pm 51)SAE \ cm^{-3}/Mm^{-1}$ and $a_1 \approx (286 \pm 46)SAE \ cm^{-3}/Mm^{-1}$,
respectively (Fig. 7). The uncertainties are large, but the main point is that the correlations show
that $a_1$ and thus $N_{CCN}(AOP_2)$ is higher for higher values of SAE. If we consider the $a_1$ values in
Table 4 as the accurate station-specific values, then using $a_1 = 286 \cdot SAE$ overestimates or
underestimates $a_1$ by +37%, +30%, -20%, -32%, -20% and +251% for SMEAR II, SORPES,
PGH, PVC, MAO and ASI, respectively. These values were calculated from $100\%(286 \cdot SAE -$
$a_1)/a_1$. The effect of the biases of $a_1$ to the biases of $N_{CCN}(AOP_2)$ are discussed in more detail in
the supplement S6. Nevertheless, we found that SAE is the only parameter that is positively
correlated with $a_1$ and that can easily be obtained from nephelometer measurements. Searching
for a more suitable proxy for $a_1$ would be an important part of follow up studies.
$R_{min}$ of (9) was estimated by calculating the 1$^{st}$ percentile of $R_{CCN/\sigma}$ at each site at each SS. The
average and standard deviation of $R_{min}$ was $5.2 \pm 3.3$ cm$^{-3}$/Mm$^{-1}$. Consequently, the
parameterization becomes
$$N_{CCN}(AOP_2) \approx \left( (286 \pm 46)SAE \cdot \ln\left( \frac{SS}{0.093 \pm 0.006} \right)(BSF - BSF_{min}) + (5.2 \pm 3.3) \right) \sigma_{sp} \qquad (10)$$
The parameterization suggests that at any supersaturation and constant scattering coefficient,
$N_{CCN}$ is the higher the smaller the particles are because both SAE and BSF are roughly inversely
correlated with the particle size. A qualitative explanation to this is that to keep $\sigma_{sp}$ constant
even if the dominating particle size decreases – which means that both SAE and BSF increase
– the number of particles has to increase. The analysis also shows that neither SAE nor BSF
alone is enough for obtaining a good estimate of $N_{CCN}$ from AOP measurements. This is again
in line with the model study of Collaud Coen et al. (2007) which showed that SAE and BSF are
sensitive to variations in somewhat different size ranges.
The parameterization in *Eq* (10) was applied to the data of the 6 stations and $N_{CCN}(AOP_2)$ was
compared with the $N_{CCN}(meas)$ at the supersaturations used in the respective CCN counters.
The results are presented as scatter plots of $N_{CCN}(AOP_2)$ vs. $N_{CCN}(meas)$ (Fig 8a and 8b), the
bias of the parameterization calculated as $N_{CCN}(AOP_2)/N_{CCN}(meas)$ (Fig 8c) and the squared
correlation coefficient $R^2$ of the linear regression of $N_{CCN}(AOP_2)$ vs. $N_{CCN}(meas)$ (Fig 8d). The
$N_{CCN}(AOP_2)$ values used for the statistics shown in Fig. 8 were calculated by using the SAE of
hourly-averaged scattering coefficients. The problem with that is that when SAE < 0 it is very
probable that also $N_{CCN}(AOP_2)$ is negative if BSF > BSF$_{min}$, as can be seen from Eq. (10). For
this reason the data with SAE < 0 were not used. The fraction of negative SAE hourly values

varied from 0.0% at SMEAR II and SORPES to 6% at MAO (Supplement S6, Table TS3). To reduce the number of rejected data, we also calculated $N_{CCN}(AOP_2)$ by using the site-specific median SAE shown in Table 4 and the hourly BSF values. The results are shown in the supplement S6.

At the site-specific lowest values of SS, the scatter plots of $N_{CCN}(AOP_2)$ vs. $N_{CCN}$(meas) of data from most stations clustered along the 1:1 line, but for the Himalayan site PGH the parameterization yielded significantly higher concentrations (Fig 8a). The bias varied from 0.7 to > 4 (Fig 8c) (Table TS3). At PGH at the lowest SS, the bias was > 4 but decreased to ~1.1-1.2 at SS = 0.4% and even closer to 1 at higher SS. At SS > 0.4%, the average bias varied between ~0.7 and ~1.3, which means $N_{CCN}$ was estimated with an average uncertainty of approximately 30% by using nephelometer data. For ASI the bias was higher, in the range of ~1.4 – 1.9. For the US coastal site PVC, the parameterization constantly underestimated the CCN concentrations by about 30%. Since $N_{CCN}(AOP_2) \approx (a_1 \ln(SS/0.093)(BSF - BSF_{min}) + R_{min})\sigma_{sp}$, it is obvious that biases of $a_1$ affect the bias of $N_{CCN}(AOP_2)$. As it was written above, the parameterization of $a_1 = 286 \cdot SAE$ overestimates or underestimates $a_1$. For most stations the bias of $N_{CCN}(AOP_2)$ can be explained by the bias of $a_1$: when $a_1$ is underestimated so is $N_{CCN}(AOP_2)$, and when $a_1$ is overestimated so is $N_{CCN}(AOP_2)$. A detailed analysis of the effect of the bias of $a_1$ on the bias of $N_{CCN}(AOP_2)$ is presented in the supplement S6.

The correlation coefficient of $N_{CCN}(AOP_2)$ vs. $N_{CCN}$(meas) is higher at higher CCN concentrations (not shown in the figure). One possible reason for this is that when CCN concentration is lower, the aerosol loading is usually lower and also the relative uncertainties of both $N_{CCN}$ and AOPs are higher than at high concentrations.

**4. Analyses of size distribution effects on $N_{CCN}$−AOP relationships**

Below we will first present effects of simulated size distributions on the relationships between $N_{CCN}$ and aerosol optical properties and then compare the simulations with field data.

**4.1 $N_{CCN}$−AOP relationships of simulated particle size distributions**

We generated lognormal unimodal size distributions as explained in section 2.6. GMD was given logarithmically evenly-spaced values from 50 nm to 1600 nm and GSD was given two values: 1.5 representing a relatively narrow size distribution and 2.0 a wide size distribution. We then calculated AOPs, $N_{CCN}$ and $R_{CCN/\sigma}$ for these size distributions.

The reasoning for the approach of estimating $N_{CCN}$ from $\sigma_{sp}$ and BSF can easily be explained by the qualitatively similar variations of $R_{CCN/\sigma}$ and BSF as function of GMD (Fig. 9). $R_{CCN/\sigma}$ is the highest for the smallest particles, i.e. for GMD = 50 nm and it decreases with an increasing GMD as also BSF. Note that the width of the size distribution has very strong effects on $R_{CCN/\sigma}$: for the wide size distribution it is approximately an order of magnitude lower than for the narrow size distribution. Note also that the values of $R_{CCN/\sigma}$ of the wide size distributions are plotted twice (Fig. 9a): the black symbols and line use the left axis to emphasize the big difference in the magnitudes of the wide and narrow size distributions; the red symbols and line use the right axis to show that the shape of the $R_{CCN/\sigma}$ size distribution is very similar to the one calculated for the narrow size distributions. The simulation also shows a potential source of uncertainty of the method: in the GMD range of ~500 − 800 nm, the BSF of the narrow size distribution actually increases, although very little with an increasing value of GMD (Fig. 9b). This phenomenon is due to Mie scattering and it is even stronger for single particles. When the size parameter x = $\pi D_p/\lambda$ of non-absorbing and weakly-absorbing spherical particles grows from ~3 to ~8, their BSF increases and then decreases again as can be shown by Mie modeling (Wiscombe and Grams, 1976). For the wavelength $\lambda$ = 550 nm this corresponds to a particle diameter range of ~525 to ~1400 nm.

The decrease of $R_{CCN/\sigma}$ and BSF with the increasing GMD was used for estimating particle sizes

with a stepwise linear regression. An example is given by the linear regressions of $R_{CCN/\sigma}$ vs.
BSF calculated for 5 consecutive size distributions, first for those that have their GMDs from
50 nm to 100 nm and the second for those that have their GMDs from 100 nm to 200 nm (Fig.
10). Note that it is obvious that linear regressions are applicable for short intervals but not well
for the whole size range. It is also obvious that an exponential fit would be perfect to explain
the relationship between $R_{CCN/\sigma}$ and BSF. But this is not what we are looking for. We are
looking for the slopes and offsets in the relationship $R_{CCN/\sigma} = aBSF + b$ that was used for fitting
the field measurement data. So, physically it would mean that $N_{CCN}$ would increase linearly as
a function of BSF even though this is not exactly correct.
The absolute values of the slopes and offsets are clearly lower for the larger particle size range.
Here, we define the particle size used for describing the size range of each regression as the
equivalent geometric mean diameter $GMD_e$, the geometric mean of the range of the GMDs of
the unimodal size distributions used for each regression. In other words,
$GMD_e = \sqrt{GMD_{low}GMD_{high}}$ , where $GMD_{low}$ and $GMD_{high}$ are the smallest GMD and the largest
GMD of the range, respectively. Two examples of the regressions were given above, one
calculated for the GMD range from 50 nm to 100 nm and the other for the GMD range from
100 nm to 200 nm. The $GMD_e$s of these two size ranges are 70.7 nm and 141.4 nm, respectively.
It will be shown below that $GMD_e$ is a mathematical concept that helps to explain the observed
relationships, not an actual GMD of the particle size distribution at the sites.
For a wide size distribution, the slopes and offsets of the regressions of $R_{CCN/\sigma}$ vs. BSF decrease
and increase, respectively, monotonically with an increasing value of $GMD_e$ in the whole size
range studied here (Fig. 11). For a narrow size distribution, the slope decreases until $GMD_e \approx$
300 nm and then increases, which means that there is no unambiguous relationship between
them. The reason is, as discussed above related to Fig 9b, that in the GMD range of ~500 – 800
nm the BSF of narrow size distributions increases slightly with an increasing GMD.
Note also that the ranges of the absolute values of the slopes and offsets of the narrow and wide
size distributions are very different. For instance, when $GMD_e = 100$ nm the slope a $\approx 4000$
cm$^{-3}$/Mm$^{-1}$ and a $\approx 1600$ cm$^{-3}$/Mm$^{-1}$ for the narrow and wide size distribution, respectively.
Since $N_{CCN}(AOP) = R_{CCN/\sigma} \cdot \sigma_{sp} = (aBSF + b)\sigma_{sp}$ this means that the $N_{CCN}(AOP)$ of narrow size
distributions is more sensitive to variations in mean particle size than the $N_{CCN}(AOP)$ of wide
size distributions.
We plotted the offset vs. slope of the unimodal size distributions and those obtained from the
linear regressions of the field data at the supersaturatios presented in Table 3 and below it the
$GMD_e$ vs. the slopes of the regressions of the unimodal size distributions (Fig 12). In Fig. 12
also the effect of the choice of the activation diameters of 50 nm, 80 nm, 110 nm, and 150 nm
is shown.
Several observations can be made in Fig. 12. First, for the simulated wide size distributions the
relationship between the offset and slope is unambiguous, while this is not the case for the
narrow size distributions at sizes $GMD_e > \sim 200$ nm (Fig 12b). Second, the field data points
roughly follow the lines of the simulations. This suggests that the slopes and offsets of the linear
regressions of $R_{CCN/\sigma}$ vs. BSF yield information on the dominating particle sizes just as they do
for the simulated size distributions. For instance, the PVC data point corresponding to the
highest supersaturation has the highest slope (1970 cm$^{-3}$/Mm$^{-3}$, Table 3) and it is close to the
wide size distribution line with the activation diameter of 50 nm (Fig. 12a). This corresponds
to the $GMD_e$ of $\sim 150$ nm (Fig. 12b). The SMEAR II high SS offset vs. slope fits best with the
corresponding lines of the narrow unimodal size distributions with activation diameters in the
range of $\sim 50 - 110$ nm and the corresponding $GMD_e \approx 150 - 200$ nm.
At the lowest SS, the offset vs. slope points of all stations agree well with the lines derived from
the unimodal modes. This is actually in line with the higher correlation coefficients ($R^2$) of the
regressions of $N_{CCN}(AOP_1)$ vs. $N_{CCN}$ (meas) at the lowest SS (Fig. 4). This can be explained by
that at low SS small particles do not get activated and unimodal size distributions in the
accumulation mode are mainly responsible for CCN. For ASI the slopes and offsets of the
lowest and highest SS are especially close to each other, closer than at any other station (Fig.
12a), and the corresponding $GMD_e \approx 750$ nm and $400$ nm, respectively, when the $GMD_e$ vs. a
relationship of any of the distributions is used (Fig. 12b). This is in line with that ASI is an
island site dominated by marine aerosols. For PGH at the lowest SS, the slope is actually
negative which is not obtained from the simulations at all so no $GMD_e$ can be given for it.
**4.2 Aerosol size characteristics of the sites**
As it was shown above, particle size distributions affect the relationships between $N_{CCN}$ and
AOPs. It is therefore discussed here how the size distributions vary at the six sites of the study
and whether they support the interpretations presented above. The size distributions are
discussed using the particle number size distribution data and the ratios of $\sigma_{sp}$ of $PM_1$ and $PM_{10}$
size ranges from those stations where they are available.
**4.2.1 Diurnal variation of particle number size distribution**
Fig. 13a shows the averaged diurnal cycle of PNSD at the sites where either a DMPS or SMPS
is available. New particle formation (NPF) events are a significant source of uncertainty in the
prediction of $N_{CCN}$ (Kerminen et al., 2012; Ma et al., 2016). Complete NPF events start from a
burst of sub-10 nm particles followed by a continuous growth up to a few hundred nanometers.
As a result, the size distribution varies significantly. NPF is one possible explanation of the
poor $N_{CCN}$-$\sigma_{sp}$ correlation.
SMEAR II and SORPES are reported to have an appreciable frequency of NPF (Kulmala et al.,
2004; Dal Maso et al., 2005; Sihto et al., 2006; Qi et al., 2015). A continuous growth of particle
size at SORPES can usually last for several days after NPF (Shen et al. 2018). Similar growth
patterns have also been observed in the Two-Column Aerosol Project (TCAP;
http://campaign.arm.gov/tcap/; refers as PVC in this study) according to Kassianov et al. (2014).
NPF is rarely observed in the Amazon forest, as reported by Wang et al. (2016). However, it
does take place at MAO as is shown in the diurnal cycle of PNSD. The reason is probably that
the MAO site was measuring aerosol downwind of the City Manaus. At ASI, there no evidence
of NPF according to the PNSD diurnal cycle.
These observations of the NPF are compared with the bias and correlation coefficients of the
parameterization discussed in section 4.1 (Fig. 8). The correlation coefficient of $N_{CCN}$ (AOP$_2$)
vs. $N_{CCN}$ (meas) is the highest, $R^2 > 0.85$ at all SS at ASI where no NPF takes place and clearly
lower at the other sites (Fig 8d). For the bias NPF appears not to have a clear influence: for both
SMEAR II and SORPES bias varies from ~1.1 to ~1.4 at SS > 0.1%. As it was stated above
(section 3.2), for most stations the bias of $N_{CCN}$(AOP$_2$) can be explained by the bias of $a_1$ in
$N_{CCN}(AOP_2) \approx (a_1 \ln(SS/0.093)(BSF - BSF_{min}) + R_{min})\sigma_{sp}$.
**4.2.2 Distributions of geometric mean diameters**
Figure 13b presents the normalized distributions of the geometric mean diameters at SMEAR
II, SORPES, PVC, MAO and ASI. They vary from 20 nm to 200 nm at all sites, with the most
frequent GMD between ~70 nm and ~120 nm depending on the site. This shows clearly that
the above-presented equivalent geometric mean diameter GMD$_e$ calculated assuming a
unimodal size distribution is not a quantitative GMD of the size distribution, it is a mathematical
concept that explains partially the relationships of $R_{CCN/\sigma}$ and BSF. However, the GMD of the
measured size distribution and GMD$_e$ are not quite comparable also for another reason. The
simulations were made by using unimodal size distributions, so that GMDe varied in the range
70 nm – 1100 nm (Fig. 11) while the GMDs were calculated from DMPS and SMPS data that
also contained the nucleation and Aitken modes that often dominate the total particle number
concentration.
The frequency distribution of GMD at SMEAR II is the widest among the five sites with PNSD
data available, followed by SORPES and PVC. At MAO the frequency distribution of GMD
has two peaks in this study. The lower peak is possibly due to the burst of sub-20 nm particles,
since these these particles have little chance to grow to sizes where they can serve as CCN. The
second peak at around 100 nm possibly represents the GMD without the burst of sub-20 nm
particles and it is distinctly narrower than at SMEAR II, SORPES and PVC.
A comparison of the correlation coefficients of $N_{CCN}$ (AOP$_2$) vs. $N_{CCN}$ (meas) (Fig. 8d) and the
widths of the GMD frequency distributions (Fig. 13b) do not show any clear relationships,
except in ASI. The frequency distribution of GMD is the narrowest at ASI, indicating that the
average particle size does not change much throughout the whole period. This is in line with
the low variation of the slope and offset of the $R_{CCN}$ vs BSF of ASI (Fig 12a). At ASI also the
correlation coefficient of $N_{CCN}$ (AOP$_2$) vs. $N_{CCN}$ (meas) is the highest, $R^2 \approx 0.8$ at all SS.
**4.3.3 Contribution of light scattering by sub-μm particles**
There is one more measure related to particle size distribution, the ratio between $\sigma_{sp}$ of sub-1
μm and sub-10 μm aerosol ($\sigma_{sp}(PM_1)/\sigma_{sp}(PM_{10})$). At SMEAR II, the contribution of submicron
particles usually varies within a range of ~0.8~0.9 and it is the highest among all sites in this
study. PVC has two peaks in the $\sigma_{sp}(PM_1)/\sigma_{sp}(PM_{10})$ distribution, the peak around 0.2
corresponding to air masses from the sea, with a very low scattering coefficient and $N_{CCN}$. By
ignoring the cleanest air masses ($\sigma_{sp}<5$ Mm$^{-1}$), the fraction of $\sigma_{sp}(PM_1)/\sigma_{sp}(PM_{10})$ is usually
around 0.8, which is just slightly lower than at SMEAR II. At PGH and MAO, the distribution
of the ratio is wider, and the peak position is at about 0.65. The overall contribution of sub-μm
particle light scattering at PGH is moderate among the sites in this study. At ASI,
$\sigma_{sp}(PM_1)/\sigma_{sp}(PM_{10})$ is the lowest among all sites in this study, indicating that particles larger
than 1 μm contribute a considerable fraction of total light scattering. For SORPES
$\sigma_{sp}(PM_1)/\sigma_{sp}(PM_{10})$ is not available.
Among those five sites, when $\sigma_{sp}(PM_1)/\sigma_{sp}(PM_{10})$ decreases, the correlation between BSF and
$R_{CCN/\sigma}$ decreases (not shown in a scatter plot). At some sites (e.g., ASI) the BSF of PM$_{10}$ is often
even larger than that of PM$_1$, which can be an error in the measurements but it may also be due
to a real phenomenon. As discussed in section 4.1, for single spherical particles Mie modeling
shows that in the particle diameter range of ~525 to ~1400 nm BSF increases with an increasing
D$_p$. Mugnai and Wiscombe (1986) simulated scattering by non-spherical particles and found
that BSF increases when the size parameter x grows from ~8 to ~15, which corresponds to the
particle diameter range of ~1400 nm to ~2600 nm at $\lambda = 550$ nm. Therefore it is obvious that
large and non-spherical particles like sea salt and dust will blur the correlation between BSF
and $R_{CCN/\sigma}$. In such a case the increase in the amount of large particles sometimes leads to an

increase of BSF and a decrease of $R_{CCN/\sigma}$, which is opposite to the usual positive correlation

between BSF and $R_{CCN/\sigma}$ in this study. This may be at least part of the explanation of the highest

bias at high values of SS in ASI (Fig 8c), the site dominated by marine aerosol. Thus, the lower

$\sigma_{sp}(PM_1)/\sigma_{sp}(PM_{10})$ may in principle result in a poor performance of our method. However, a

comparison of the correlation coefficients and the $\sigma_{sp}(PM_1)/\sigma_{sp}(PM_{10})$ frequency distributions

of each site shows the opposite. At the highest SS of each site, the $R^2$ in a decreasing order is

ASI, PGH, MAO, SORPES, SMEAR II, and PVC (Fig. 8d). The peaks, i.e. modes of the

frequency distribution of $\sigma_{sp}(PM_1)/\sigma_{sp}(PM_{10})$ are, in a growing order, ASI: 0.375, PGH: 0.625,

MAO: 0.65, PVC: 0.825, SMEAR II: 0.875. Note that at SORPES there is only one size range

measured. Of these the $R^2$ of only PVC and SMEAR II are not in the same order (Fig 8d). This

suggests that $N_{CCN}$ can be estimated better from the aerosol optical properties for sites

dominated by large particles than for sites dominated by small particles. This further suggests

that the ambient size distributions were so wide that the non-monotonous relationship between

particle size and BSF discussed above did not play an important role. On the other hand, the

bias at the highest SS has no clear relationship with $\sigma_{sp}(PM_1)/\sigma_{sp}(PM_{10})$. .

There is also an additional observation that can be made. The above-mentioned order of the

modes of the frequency distribution of $\sigma_{sp}(PM_1)/\sigma_{sp}(PM_{10})$ is almost the same as the order of

the slopes and offsets and $GMD_e$s in Fig. 12. Only for SMEAR II and PVC the order is not the

same. This further supports the interpretation that the slopes and offsets of the linear regression

of $R_{CCN}$ vs BSF depend on the dominating particle size of particle size distribution.

**5. Conclusions**

The relationships between aerosol optical properties, CCN number concentrations ($N_{CCN}$) and

particle number size distributions were investigated based on in-situ measurement data from

six stations in very different environments around the world. The goals were to find physical

explanations of the relationships and to find a parametrization to obtain $N_{CCN}$ from sites where

AOPs are measured but no CCN counter is available. There are many previous

parameterizations for doing just the same. As a starting point we used the parameterization
presented by Jefferson (2010). That one needs also absorption measurements since it includes
single-scattering albedo. We instead studied how the parameterization would look like if only
total scattering and backscattering data were available.
The basic idea for the parameterization is that $N_{CCN}$ is proportional to $\sigma_{sp}$ and a function of the
backscatter fraction (BSF), i.e., $N_{CCN}(AOP) = (aBSF + b)\sigma_{sp}$ as is also in the parameterization
of Jefferson (2010). In the study of the physical explanation of the relationships between $N_{CCN}$
and AOPs, we found that the slope a and offset b in $N_{CCN}(AOP) = (aBSF + b)\sigma_{sp}$ depend clearly
on the dominating particle size and on the width of the size distributions. This was shown first
by simulations and then by comparisons of the simulations with field data. The analyses showed
that the sensitivity of $N_{CCN}(AOP)$ to variations of BSF increases with a decreasing particle size.
As a result, sites dominated by supermicron aerosol particles, such as ASI that is dominated by
marine aerosol, have a small value of the slope a in the above formula, which means that it is
not very sensitive to variations in BSF. Sites dominated by small aerosol particles are clearly
more sensitive. For instance for the coastal site PVC that is significantly affected by
anthropogenic emissions, the slope a in the above formula is an order of magnitude higher than
at the marine site.
A logarithmic function was fitted to the $N_{CCN}$ vs. supersaturation SS data in the range SS < 1.1%.
For $N_{CCN}(AOP)$ the fitting yielded a logarithmic dependence on SS: $N_{CCN}(AOP) \approx$
$(286 \cdot SAE \cdot \ln(SS/0.093)(BSF - BSF_{min}) + (5.2 \pm 3.3))\sigma_{sp}$. Actually this result is qualitatively in
line with the relationship between AOD and CCN reported by Andreae (2010). The derived
$N_{CCN}(AOP)$ depends on $\sigma_{sp}$, SAE and BSF. The analysis shows that neither SAE nor BSF alone
is enough for obtaining a good estimate of $N_{CCN}$ from AOP measurements.          .
At the lowest supersaturations of each site (SS $\approx$ 0.1%), the average bias, defined as the ratio
of the AOP-derived and measured $N_{CCN}$, varied from ~0.7 to ~1.9 at most sites except at the
Himalayan site PGH where the bias was > 4. At SS > 0.4% the average bias ranged from ~0.7
to ~1.3 at most sites. For the marine-aerosol dominated site ASI the bias was higher, ~1.4 – 1.9.

In other words, at SS > 0.4% $N_{CCN}$ was estimated with an average uncertainty of approximately 30% by using nephelometer data. The biases were mainly due to the biases in the parameterization related to the scattering Ångström exponent SAE.

**Author contributions**

YS carried out measurements at SORPES in China, analyzed and visualized data of all sites, and wrote the original draft. AV contributed to data analysis and visualization, writing and editing the original draft, and supervised the work of YS in Finland. AD provided funding for the measurements and research at SORPES in China, acquired funding for YS in China, and supervised the work of YS. KL, HK and PA carried out measurements, data collection and maintenance of measurement data of SMEAR II in Finland. YS, XC, XQ, WN and XH carried out measurements, data collection and maintenance of measurement data of SORPES in China. MK and TP provided the funding for YS in Finland. MK provided funding for the measurements and research at SMEAR II in Finland. TP and VMK formulated the goals of the research and supervised it.

**Acknowledgments**

This work was supported by the National Key Research & Development Program of Ministry of Science & Technology (MOST) of China (2016YFC0202000, 2016YFC0200500), Academy of Finland via Center of Excellence in Atmospheric Sciences (project no. 272041), National Natural Science Foundation of China (41725020, 91544231) and the Collaborative Innovation Center of Climate Change supported by the Jiangsu 2011 Program. Scholarship for Y.Shen was supported by both China Scholarship Council (CSC) and the Centre for International Mobility (CIMO) of Finland. Data were also obtained from the Atmospheric Radiation Measurement (ARM) User Facility, a U.S. Department of Energy (DOE) Office of Science user facility managed by the Office of Biological and Environmental Research. For the SMEAR II data we thank the SMEARII technical team.

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

**Tables**
Table 1. Site and data description

| Dataset | Description | Location | Period | CCN | | Size distrubtion | | AOPs | |
|---|---|---|---|---|---|---|---|---|---|
| | | | | Instrument | SS | Instrument | size range | Instrument | inlet |
| SMEAR II | Boreal Forest, Finland | 61°51' N, 24°17' E, 179m | 2016.1.1-2016.12.31 | CCN-100 | 0.1%, 0.2%, 0.5% and 1.0% | DMPS custom-made | 3-1000nm | Nephelometer TSI-3563 | PM1, PM10 |
| SORPES | Urban agglomerations, China | 32°07' N, 118°56'E, 40m | 2016.06.01-2017.05.31 | CCN-200 | 0.1%, 0.2%, 04%, and 0.8% | DMPS custom-made | 6-800nm | Nephelometer Aurora-3000 | TSP |
| PGH[a] | Ganges Valley, India | 29°22' N, 79°27' E, 1936m | 2011.11.01-2013.03.25 | CCN-100 | 0.12%, 0.22%, 0.48% and 0.78% | NA | NA | Nephelometer TSI-3563 | PM1, PM10 |
| PVC[b] | Cape Cod, USA | 42°2' N, 70°3' W, 43m | 2012.07.16-2012.09.30 | CCN-100 | 0.15%, 0.25%, 0.4% and 1.0% | SMPS TSI-3936 | 11-465nm* | Nephelometer TSI-3563 | PM1, PM10 |
| MAO[c] | Downwind Manaus City, Brazil | 3°13' S, 60°36 W, 50m | 2014.01.29-2014.12.31 | CCN-100 | 0.25%, 0.4%, 0.6% and 1.1% | SMPS TSI-3936 | 11-465nm* | Nephelometer TSI-3563 | PM1, PM10 |
| ASI[d] | Ascension Island, Atlantic Ocean | 7°58' S, 14°21' W, 341m | 2016.06.01-2017.10.19 | CCN-100 | 0.1%, 0.2%, 0.4%, and 0.8% | SMPS TSI-3936 | 11-465nm* | Nephelometer TSI-3563 | PM1, PM10 |

[a] use products: aipavg1ogrenM1.c1., and aosccnavgM1.c2.

[b] use products: aipavg1ogrenM1.s1., noaaaosccn100M1.b1., and aossmpsS1.a1.

[c] use products: aip1ogrenM1.c1., aosccn1colM1.b1., and aossmpsS1.a1.

[d] use products: aosnephdryM1.b1., aosccn2colaavgM1.b1., and aossmpsM1.a1.

* vary slightly
Table 2. Descriptive statistics of AOPs of PM10 aerosol and $N_{CCN}$ at the different sites. $\sigma_{sp}$:
total scattering coefficient of green light ($\lambda = 550$ nm or 525 nm), in Mm$^{-1}$; BSF: backscatter
fraction of green light; SAE: scattering Ångström exponent between blue and red light. The
$N_{CCN}$ statistics in # cm$^{-3}$ are presented for four supersaturations (SS) at each site. The numbers
are the averages and standard deviations.

| | | AOPs | | | CCN | | | |
|---|---|---|---|---|---|---|---|---|
| | $\sigma_{sp}$ | BSF | SAE | | #1 | #2 | #3 | #4 |
| SMEAR II | 14±14 | 0.15±0.03 | 2.11±0.67 | SS: | 0.10% | 0.20% | 0.50% | 1.00% |
| | | | | $N_{CCN}$: | 129±99 | 303±229 | 514±388 | 740±511 |
| SORPES | 270±188 | 0.11±0.02 | 1.45±0.33 | SS: | 0.10% | 0.20% | 0.40% | 0.80% |
| | | | | $N_{CCN}$: | 974±632 | 2377±1244 | 4199±1915 | 5363±2245 |
| PGH | 239±215 | 0.07±0.01 | 0.53±0.30 | SS: | 0.12% | 0.22% | 0.48% | 0.78% |
| | | | | $N_{CCN}$: | 325±296 | 935±621 | 2359±1391 | 2882±1707 |
| PVC | 27±22 | 0.13±0.03 | 1.79±0.52 | SS: | 0.15% | 0.25% | 0.40% | 1.00% |
| | | | | $N_{CCN}$: | 515±361 | 864±603 | 1163±774 | 1766±1020 |
| MAO | 24±19 | 0.14±0.02 | 1.00±0.55 | SS: | 0.25% | 0.40% | 0.60% | 1.10% |
| | | | | $N_{CCN}$: | 448±377 | 783±693 | 1034±923 | 1251±1068 |
| ASI | 20±13 | 0.14±0.01 | 0.73±0.41 | SS: | 0.10% | 0.20% | 0.40% | 0.80% |
| | | | | $N_{CCN}$: | 113±79 | 234±175 | 271±199 | 319±203 |

Table 3. The slopes and offsets of ordinary linear regressions of $R_{CCN/\sigma}$ vs. BSF at the different

supersaturation SS at the studied sites. s.e.: standard error of the respective coefficient obtained

from the linear regressions. The unit of the coefficients is $[N_{CCN}]/[\sigma_{sp}]$ =cm$^{-3}$/Mm$^{-1}$.

| | | $R_{CCN/s}$ = aBSF + b | |
|---|---|---|---|
| | SS (%) | a ± s.e. | b ± s.e. |
| SMEAR II | 0.10 | 91 ± 3 | -2.9 ± 0.4 |
| | 0.20 | 433 ± 5 | -38.6 ± 0.7 |
| | 0.50 | 867 ± 10 | -86.4 ± 1.5 |
| | 1.00 | 1155 ± 17 | -115.8 ± 2.5 |
| SORPES | 0.10 | 62 ± 2 | -2.6 ± 0.2 |
| | 0.20 | 266 ± 4 | -18.4 ± 0.4 |
| | 0.40 | 531 ± 7 | -39.1 ± 0.8 |
| | 0.80 | 738 ± 11 | -55.9 ± 1.2 |
| PGH | 0.12 | -18 ± 1 | 2.6 ± 0.1 |
| | 0.22 | 24 ± 3 | 2.8 ± 0.2 |
| | 0.48 | 244 ± 12 | -4.4 ± 0.8 |
| | 0.78 | 344 ± 14 | -8.3 ± 1.0 |
| PVC | 0.15 | 417 ± 9 | -30.2 ± 1.1 |
| | 0.25 | 793 ± 17 | -61.7 ± 2.1 |
| | 0.40 | 1176 ± 25 | -95.3 ± 3.1 |
| | 1.00 | 1945 ± 43 | -161.4 ± 5.3 |
| MAO | 0.25 | 273 ± 5 | -19.0 ± 0.7 |
| | 0.40 | 544 ± 8 | -42.9 ± 1.2 |
| | 0.60 | 678 ± 13 | -50.9 ± 1.8 |
| | 1.10 | 868 ± 32 | -58.3 ± 4.3 |
| ASI | 0.10 | 22 ± 2 | 2.2 ± 0.2 |
| | 0.20 | 105 ± 3 | -3.6 ± 0.5 |
| | 0.40 | 127 ± 4 | -5.0 ± 0.6 |
| | 0.80 | 136 ± 4 | -4.0 ± 0.6 |

Table 4. The coefficients $a_0$, $a_1$, $b_0$ and $b_1$ obtained from the fitting of $a = a_1\ln(SS) + a_0$ and $b =$

$b_1\ln(SS) + b_0$ with the data in Table 3. The unit of the coefficients is $[N_{CCN}]/[\sigma_{sp}]$ =cm$^{-3}$/Mm$^{-1}$.

s.e.: standard error of the respective coefficient obtained from the regressions. SAE: scattering

Ångström exponent of PM10 aerosol.

| | $R_{CCN/\sigma}$ = (a$_1$ln(SS) + a$_0$)BSF + b$_1$ln(SS) + b$_0$ | | | | SAE | |
|---|---|---|---|---|---|---|
| SITE | a$_1$ ± s.e. | a$_0$ ± s.e. | b$_1$ ± s.e. | b$_0$ ± s.e. | average ± std | median |
| SMEAR II | 464 ± 11 | 1170 ± 16 | -49 ± 1.5 | -118 ± 2.1 | 2.11 ± 0.67 | 2.22 |
| SORPES | 331 ± 12 | 817 ± 18 | -26 ± 0.9 | -62 ± 1.4 | 1.45 ± 0.33 | 1.50 |
| PGH | 205 ± 30 | 385 ± 41 | -6.3 ± 1.5 | -9.1 ± 2.0 | 0.53 ± 0.30 | 0.57 |
| PVC | 810 ± 17 | 1933 ± 21 | -70 ± 1.7 | -160 ± 2.1 | 1.79 ± 0.52 | 1.91 |
| MAO | 393 ± 45 | 858 ± 40 | -25 ± 6.6 | -60 ± 5.8 | 1.00 ± 0.55 | 1.09 |
| ASI | 52 ± 17 | 164 ± 26 | -2.9 ± 1.6 | -6.3 ± 2.3 | 0.73 ± 0.41 | 0.64 |

**FIGURES**

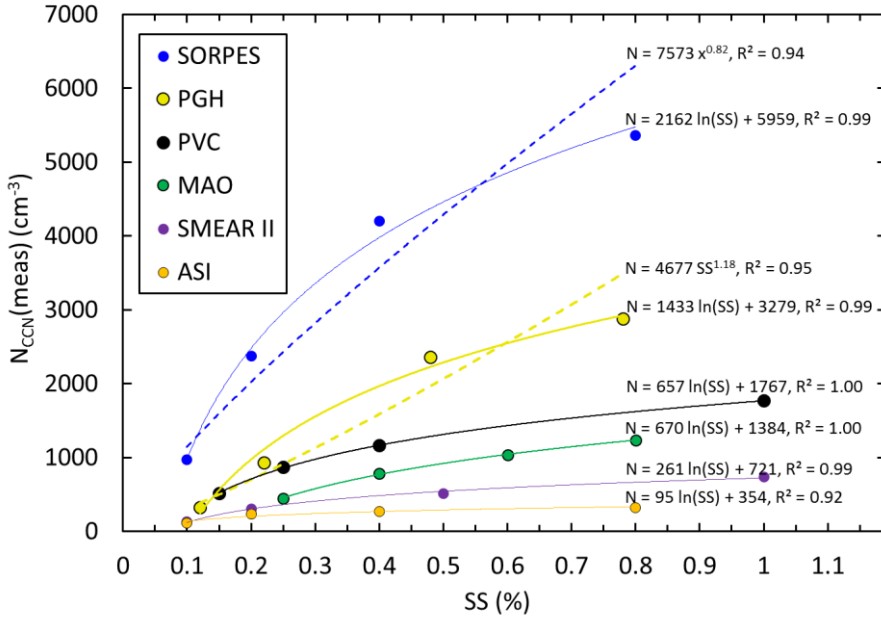

Figure 1. Averages of the measured $N_{CCN}$ at the six sites at the station-specific supersaturations
of the CCN counters and a logaritmic fitting to the data.

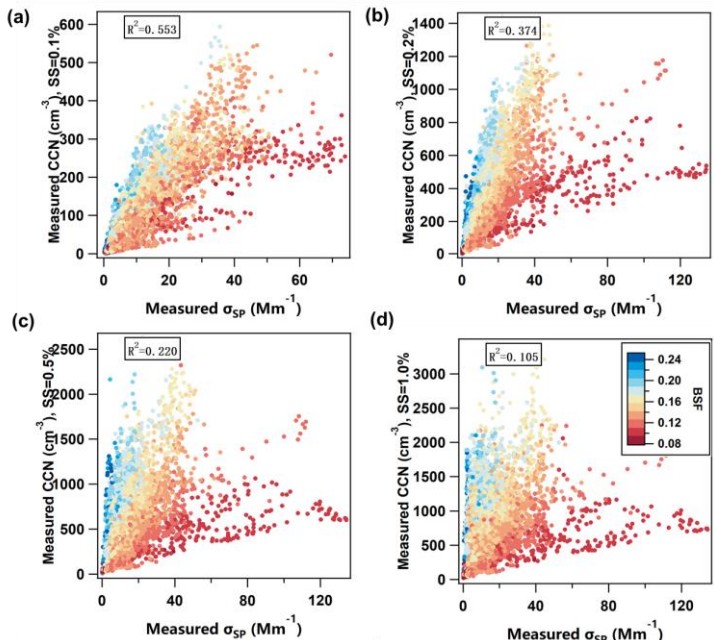

Figure 2. Measured CCN number concentration $N_{CCN}$(meas) vs. $PM_{10}$ scattering coefficient $\sigma_{sp}$
at $\lambda$ = 550 nm at SMEAR II at four supersaturations (SS): a) 0.1 %, b) 0.2 %, c) 0.5 % and d) 1.0 %.
Colorcoding: backscatter fraction (BSF) at $\lambda$ = 550 nm.

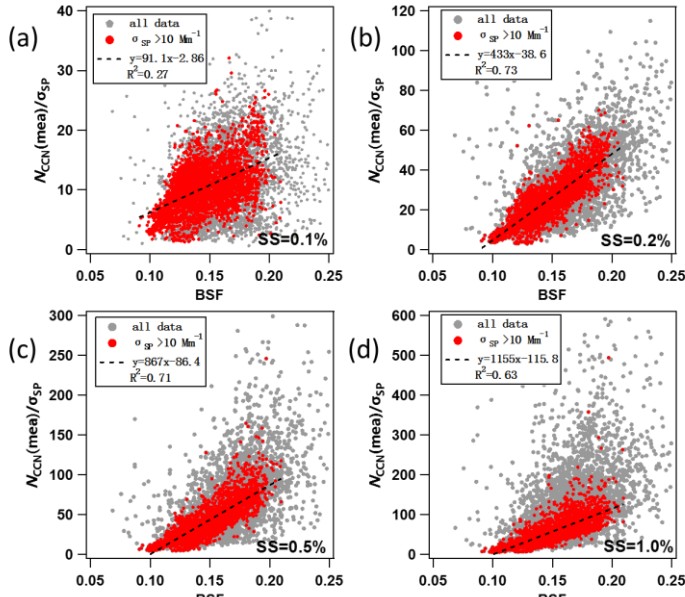

Figure 3. Relationship between $R_{CCN/\sigma}$ (= $N_{CCN}$(meas)/$\sigma_{sp}$) and BSF at SMEAR II at four
supersaturations (SS): a) 0.1 %, b) 0.2 %, c) 0.5 % and d) 1.0 %. Grey symbols: all data, red
symbols: data at $\sigma_{sp}$ > 10 Mm$^{-1}$. Both $\sigma_{sp}$ and BSF were measured at $\lambda$ = 550 nm.

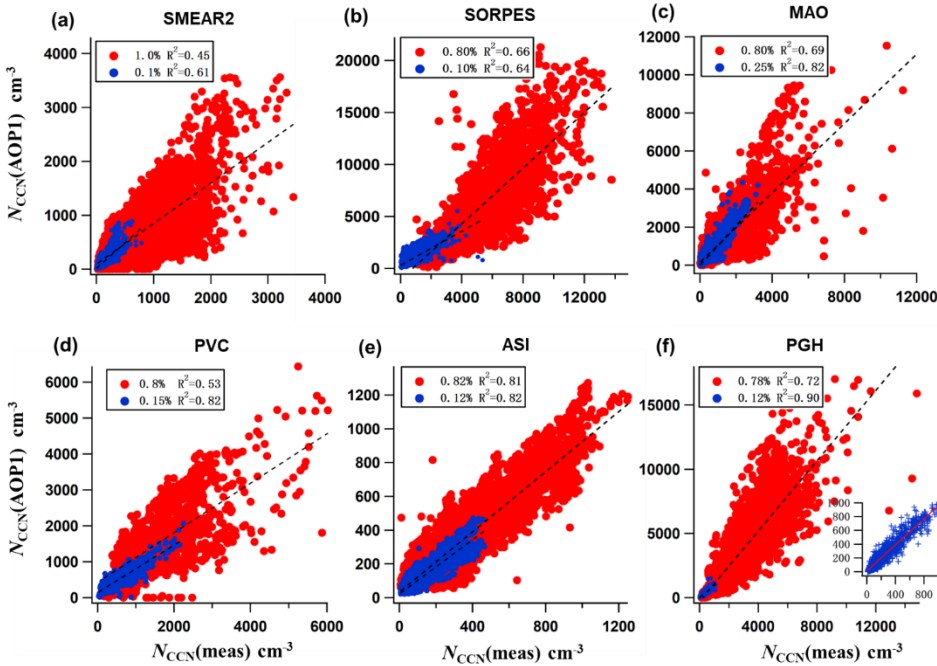

Figure 4. $N_{CCN}$ (AOP$_1$) vs. $N_{CCN}$ (meas) at a) SMEAR II, b) SORPES, c) MAO, d) PVC, e) ASI and f)
PGH. $N_{CCN}$(AOP) was calculated by using the slopes and offsets a and b of the linear regressions
$R_{CCN/\sigma}$ = aBSF + b in Table 3 for two supersaturations (blue symbols: low SS, red symbols: high
SS).

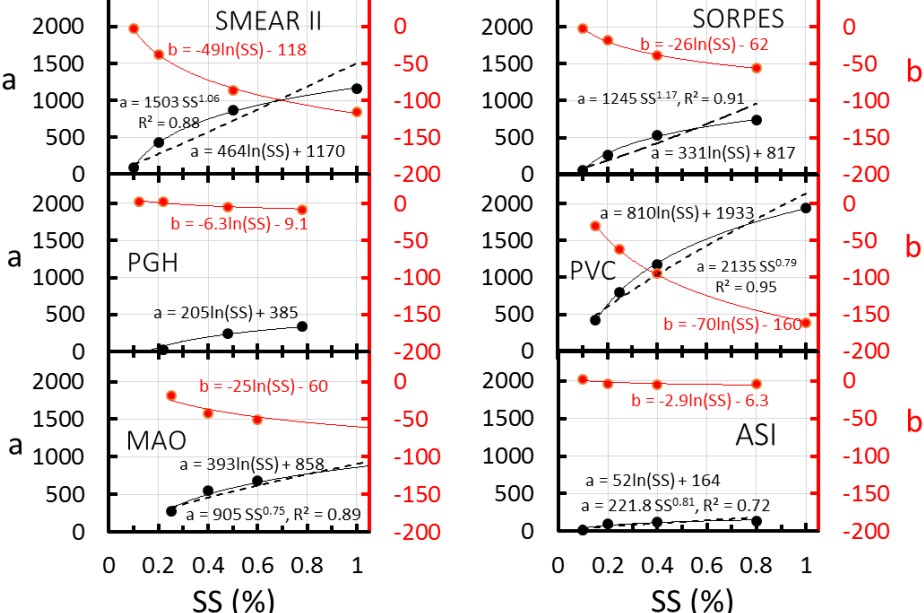

Figure 5. The the slopes and offsets a and b of the linear regressions $R_{CCN/\sigma} = aBSF + b$ of each station (Table 3) as a function of supersaturation SS. Two types of functions, a logarithmic and a power fuction were fitted to the coefficient a, to coefficient b only a logaritmic function. The squared correlation coefficients $R^2$ are shown only for the power function fittings, for the logarithmic fittings they were all > 0.99.

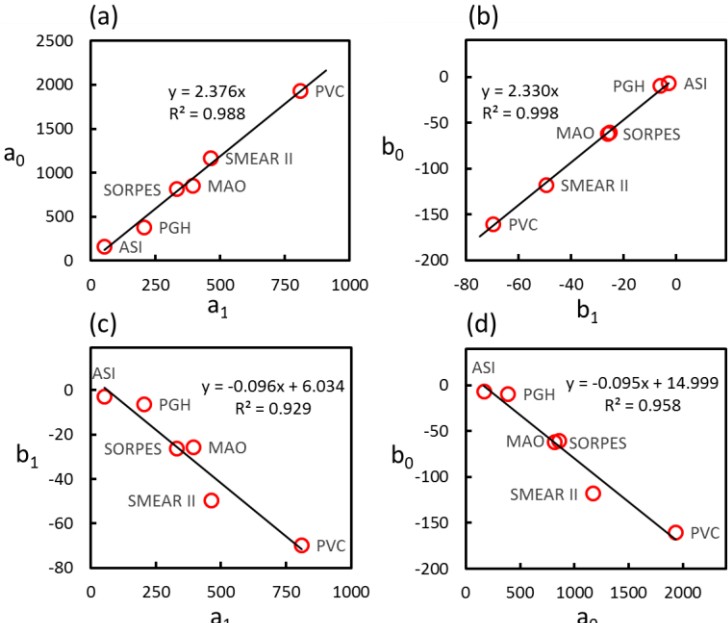

Figure 6. Relationship between the coefficients $a_0$, $a_1$, $b_0$ and $b_1$ of Equation (7) for each station presented in Table 4 for the 6 stations. a) $a_0$ vs. $a_1$, b) $b_0$ vs. $b_1$, c) $b_1$ vs. $a_1$, d) $b_0$ vs. $a_0$.

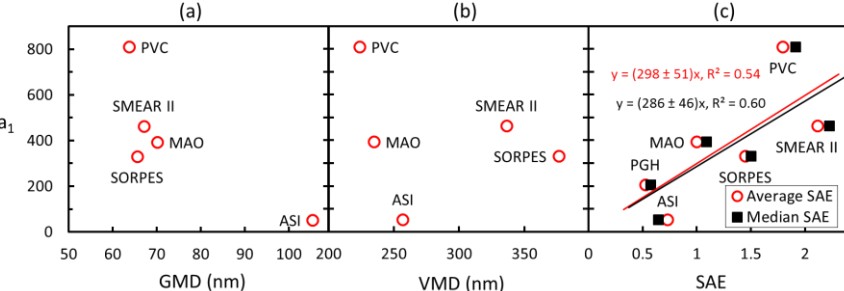

Figure 7. Relationship of the $a_1$ coefficient in Equation (8) with the average a) geometric mean diameter of the PNSD data size ranges of the sites, b) volume mean diameter of the same size range, and c) PM$_{10}$ scattering Ångström exponent (SAE).

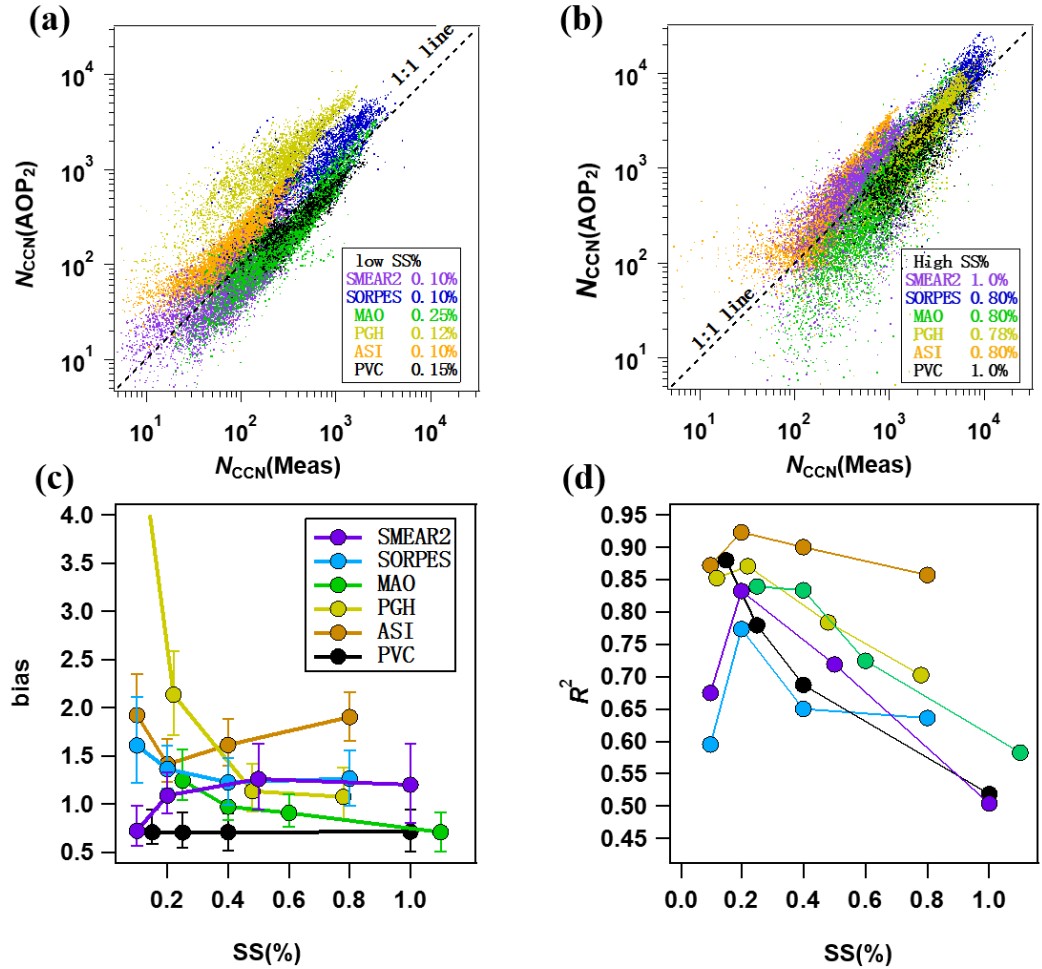

Figure 8. Statistics of $N_{CCN}(AOP_2)$ from parameterization in Eq. (10). $N_{CCN}(AOP_2)$ vs. $N_{CCN}$(meas) at different sites at relatively (a) low and (b) high supersaturations, (c) bias = $N_{CCN}(AOP_2)/N_{CCN}$ (meas) at different sites and supersaturations, and (d) $R^2$ of the linear regression of $N_{CCN}(AOP_2)$ vs. $N_{CCN}$ (meas) at different sites and supersaturations.

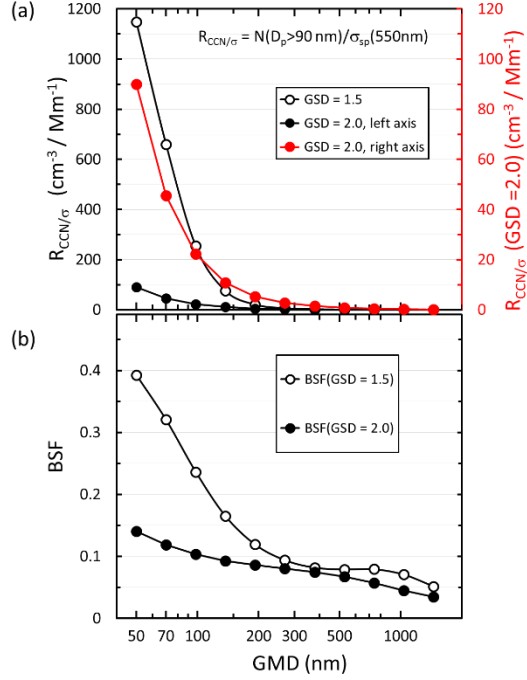

Figure 9. Size distribution of a) $R_{CCN/\sigma}$ and b) backscatter fraction BSF ($\lambda$ = 550 nm) of simulated

narrow (GSD = 1.5) and wide (GSD = 2.0) unimodal size distributions. GMD: geometric mean

diameter, GSD: geometric standard deviation. Note: in a) the $R_{CCN/\sigma}$ of the wide size

distributions are plotted twice: the black symbols and line use the left axis to emphasize the

big difference in the magnitudes of the wide and narrow size distributions; the red symbols

and line use the right axis to show that the shape of the $R_{CCN/\sigma}$ size distribution is very similar

to the one calculated for thew narrow size distributions. $R_{CCN/\sigma}$ was calculated assuming

particles larger than 90 nm get activated.

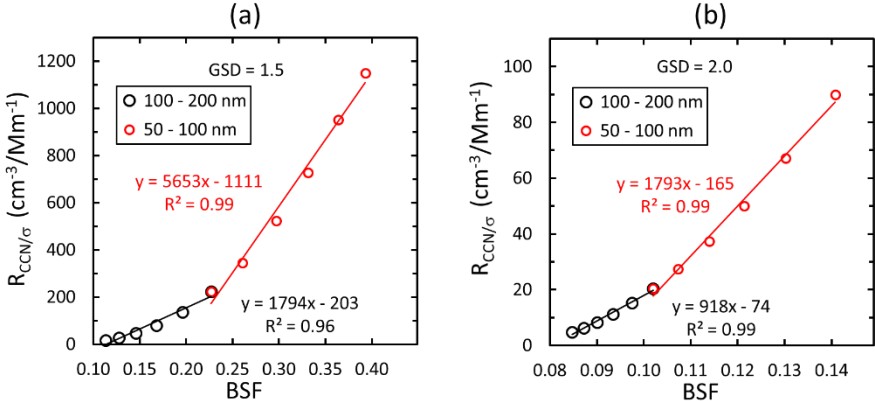

Figure 10. Linear regressions of $R_{CCN/\sigma}$ vs backscatter fraction BSF ($\lambda$ = 550 nm) of simulated

unimodal a) narrow (GSD = 1.5) and b) wide (GSD = 2.0) size distributions. The regressions

were calculated assuming that the data consist of size distributions with GMD ranging from 50

to 100 nm and 100 to 200 nm. $R_{CCN/\sigma}$ was calculated assuming particles larger than 90 nm get

activated.

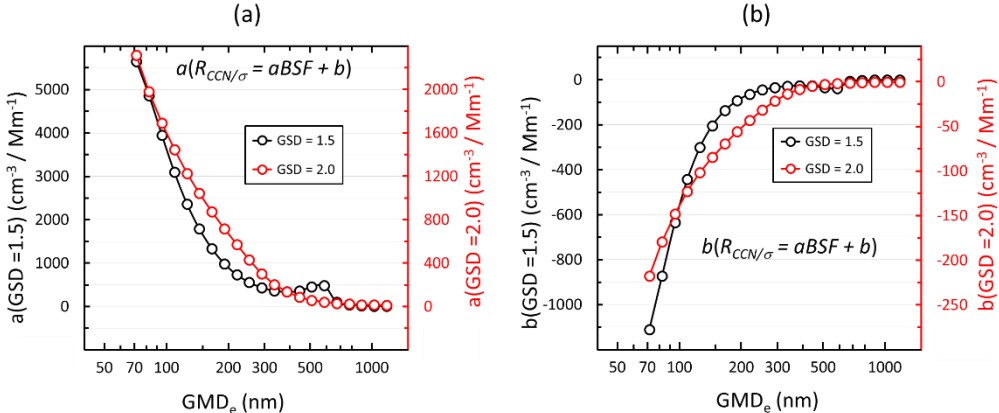

Figure 11. Size distributions of the coefficients of the linear regressions of $R_{CCN/\sigma}$($\lambda$ = 550 nm)

vs backscatter fraction BSF ($\lambda$ = 550 nm) of narrow and wide size distributions. a) slopes of

$R_{CCN/\sigma}$ vs. BSF, b) offsets of $R_{CCN/\sigma}$ vs. BSF. $R_{CCN/\sigma}$ was calculated assuming particles larger than

90 nm get activated. The regressions were calculated for 5 consequtive size distributions.

$GMD_e$ is the geometric mean of the range of the unimodal size distributions used for the

regressions.

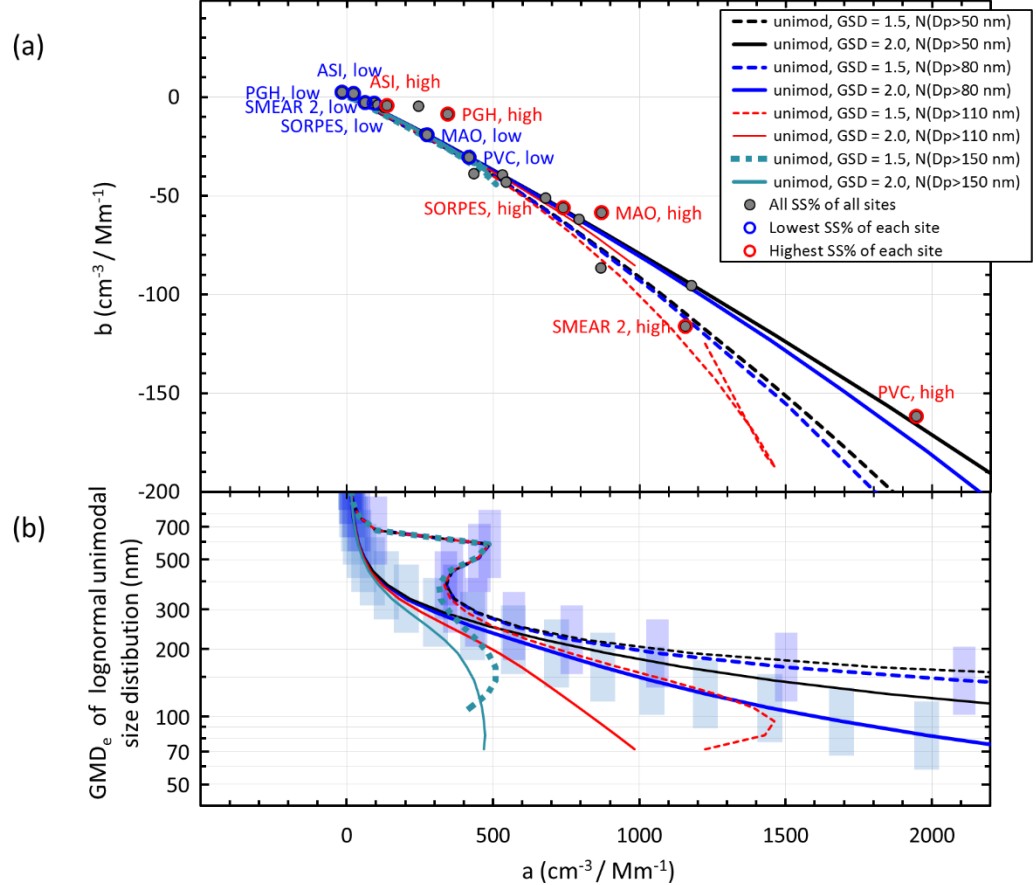

Figure 12. a) Relationships of the slopes and offsets of the linear regressions of $R_{CCN/\sigma}$ = aBSF + b vs. BSF of the simulated unimodal narrow (GSD = 1.5) and wide (GSD = 2.0) size distributions and those obtained from the similar regressions of the station data (Table 3). b) Equivalent geometric mean diameter (GMD$_e$) of the unimodal modes used for the linear regression vs. the slope of the linear regression of $R_{CCN/\sigma}$ vs. BSF. The vertical error bars show the ranges of the GMDs of the unimodal size distributions used in the respective linear regressions. $R_{CCN/\sigma}$ was calculated for the activation diameters of 50 nm, 80 nm, 110nm, and 150 nm.

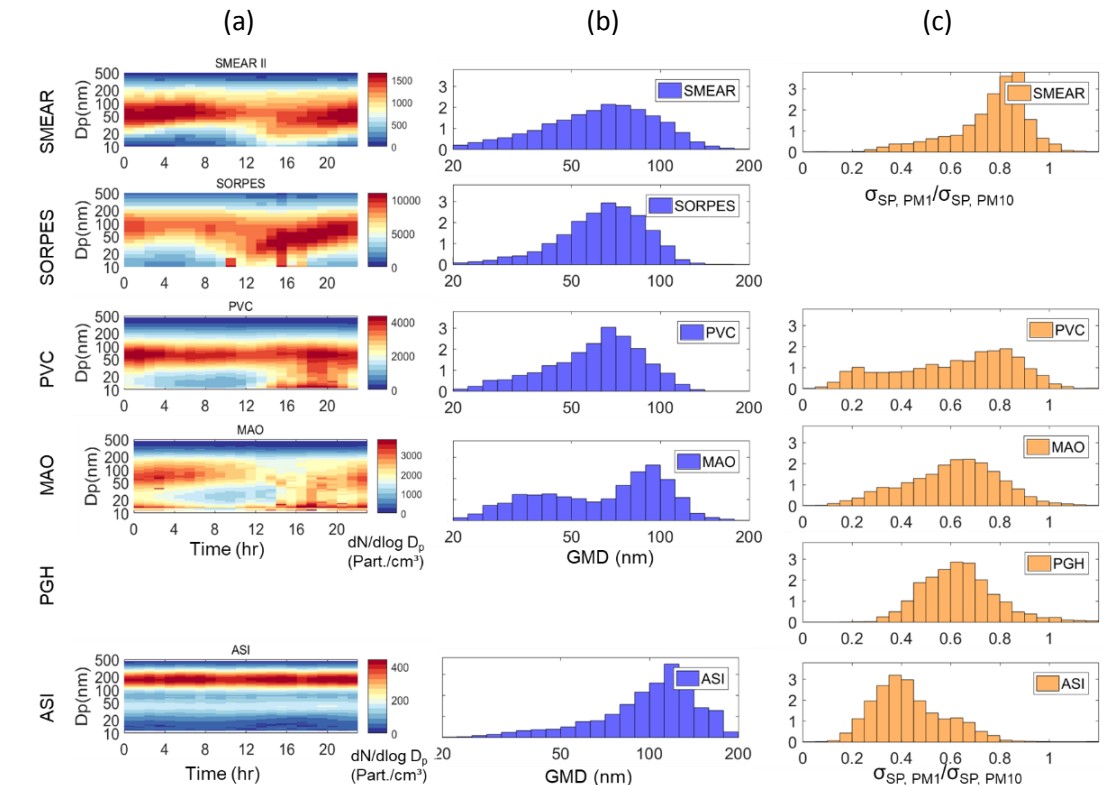

Figure 13. Analyses of particle size distributions at the six sites. a) Average diurnal cycle of PNSD and b) normalized size distribution of GMD at SMEAR II, SORPES, PVC, and ASI, c) normalized frequency distribution of $\sigma_{sp}(PM_1)/\sigma_{sp}(PM_{10})$ at SMEAR II, PVC, MAO, PGH and ASI.