# Peer review of "Estimating CCN number concentrations using aerosol optical properties: Role of particle number size distribution and parameterization"

_Atmospheric Chemistry and Physics, 2019_

## Referee Comment (RC1) · Anonymous Referee #1 · 6 May 2019

Summary:

The manuscript by Yicheng Shen et al. entitled "Estimating CCN number concentrations using aerosol optical properties: Role of particle number size distribution and parameterization" presents a para-metrization of the relationship between aerosol optical properties (AOP) and cloud condensation nu-clei number concentrations (NCCN) measured at different ground-based stations under contrasting at-mospheric conditions. The influence of different aerosol size distributions on this relationship have been tested and the authors have shown that the parametrization is mainly driven by the geometrical mean diameter and the width of the aerosol size distribution. The

parametrization can be used to estimate NCCN by AOP measurements at different sites where AOD are measured but no NCCN.

My overall recommendation is that the manuscript should be published after some major changes. Overall, this is thorough and interesting study. From a formal perspective, the quality of the manu-script is high - it is well structured and all arguments and aspects are presented clearly. From a scien-tific perspective, it shows a careful and extensive analysis with a proper physical discussion. However, some critical aspects have to be clarified before publication.

Major and General Comments:

1) Page 1 line 26 of the Abstract is misleading. The manuscript focusses on the relationship be-tween AOD and CCN, this parametrization also depends on the supersaturation, however the manuscript does not show that the relationship between measured CCN number concentra-tion and the corresponding supersaturation (S) is logarithmic, there is even no figure and no discussion about the relationship between CCN and S in the manuscript.

2) The manuscript is very technical, following the manuscript would have fit perfect in the jour-nal aerosol measurement techniques as well. Please clarify why this is an ACP rather than an AMT manuscript.

3) Table 1 it would be very helpful to add a small description of the site.

4) Page 6 line 14 the authors mention that no severe pollution episodes where observed in the data used for this study. However, table 1 shows that the data used for this study span a whole year. Was the data selected for no pollution periods and if so, what was the criterium?

5) Page 7 line 3-9 describes which data have been excluded for the analysis. It would be nice to mention the percentage of excluded data by the different criteriums and in total.

6) Page 9 line 9-11: The authors describe that NCCN was calculated by integrating the number concentration of aerosol particles larger than a certain diameter. How these diameters were chosen? Using the Köhler Theory and a certain hygroscopicity for the particles? If yes, which hygroscopicity have been used?

7) Page 10 line 28: The authors mention that the error in the measurements influence the result of the linear regression. As long as the linear relationship between measured variables which both have an uncertainty plays a key role in this study, the author should use a bivariate re-gression including the uncertainty of all measured variables as introduced by Cannell 2008.

8) Page 11 line 18: Is there also a lower R2 for higher NCCN concentrations? Figure 8 b suggest such a closure.

9) Page 12 line 12: What is a reasonable long period?

10) Page 12 line 22-23: What is the percentual uncertainty?

11) Page 14 line 6-7: The authors wants to show the different magnitude in Figure 9a, following I would recommend to use only one axis, by that the different magnitudes would be visible eas-ier.

12) Page 16 line 24: The MAO site was measuring aerosol downwind of the City Man-aus and not Amazon rainforest air. The pronounced nucleation size particles suggest predominant anthro-pogenic emissions from Manaus.

13) Page 17 line 14: The authors compares the result from MAO with the results from ATTO. The disagreement of this comparison is a strong indication that the air masses measured at MAO are not representative for the Amazon rainforest. Please clarify this aspect.

Minor Comments

- The authors always write 'SS%' instead of 'SS' or 'S' for the supersaturation. The '%'

behind the value is sometimes used, sometimes not. I would recommend to use 'SS' or 'S' for the supersaturation and always mention the unit behind a variable.

- Page 13 line 5: two times "of"

- Page 14 line 9: is this a complete sentence?

- Page 16 line 1: is the double negation on purpose?

Cantrell, C. A.: Technical Note: Review of methods for linear leastsquares fitting of data and applica-tion to atmospheric chemistry problems, Atmos. Chem. Phys., 8, 5477–5487, 12 doi:10.5194/acp-8-5477-2008, 2008.

––––––––––––––––––––––––––––––

---

## Referee Comment (RC2) · Anonymous Referee #2 · 17 May 2019

The paper is on an interesting subject and the parameterization of CCN concentrations by optical parameters can be useful. The data are carefully analyzed and the parameterization is developed step by step, which helps understanding the resulting complicated formula. Overall, I find that this is a paper that is in principle publishable after moderate to major revisions. One reservation I have, is that it is clearly a method paper that can certainly be very useful, but does not have major scientific results (as to e.g., actual CCN in various environments and larger implication), so ACP is in my opinion not the best journal for this. It fits much better in a method-oriented journal. I would strongly suggest to submit it somewhere else. However, in case it is published in ACP the following comments should be addressed:

Parts of the steps taken could be better motivated and better interpreted (see comments below), which would make the manuscript easier to follow and less technical and dry. Overall I think the length of the paper and the 13! Figures are a bit excessive. Please think of a way to summarize the information in a more compact way.

The writing needs to be improved overall and the English corrected in some places (many formulations are unclear and sometimes grammatically incorrect). While I cannot correct all of these, I will give some examples and the further corrections should not pose a big problem to some the co-authors on this paper.

Specific comments:

1) Page 3, line 4: Explain better, why they are not adequate. I assume you mean the spatial coverage? Are optical properties really more widely measured than size distributions? Support this claim!

2) Page 3, line 26: Do you mean N_CCN/sigma_sp instead of R_CCN/sigma_sp? Later you call this ratio R_(CCN/sigma) please be consistent.

3) Page 8, line 6-9: This paragraph consists of several incomplete, grammatically incorrect sentences and as a result, it is unclear what you are actually doing in this calculation. Please formulate clearly and explain in more detail.

4) Page 9, line 5: How well do you think a unimodal size distribution can describe a realistic situation, where size distributions are at least bi-modal? Give some motivation already here for choosing this approach.

5) Page 9, line 13: The section is called "Overview of measured properties", but it does not contain an overview of measured properties, more a quality control. It should be merged with the method section.

6) Page 9, line 14-24: This Paragraph seems out of place here. At the beginning of a new section, before the sub-sections, I would expect some general introduction. Instead we get some details of only one station that are neither well explained nor well

motivated. Please rearrange this section and find a better place for this information (after you clarify it).

7) Page 9, line 19: You do not make clear how you make this fit (e.g., how you determine k) and why it does not fit. Is this relevant for the paper in any way? It well known that an error function gives a much better fit to CCN spectra.

8) Page 9, line 24: From the previous discussion it is not clear, why/how this parameterization involved absorption data. If this paragraph is absolutely necessary, please explain in more detail, otherwise I would suggest to omit this discussion entirely.

9) Page 10, line 13/14: Give an interpretation of this fact and a motivation, why you investigate this dependence.

10) Page 11, line 11/12: Please use the appropriate reduced major axis regression then, instead of a regression that is not suitable for your data

11) Page 12, line 3/4: I am not sure what you mean by "underlying reason"? $b_0$ is the limiting value for SS=1 and the range of SS stays the same. So go from Nccn $\sim$ 0 at approx. SS $\sim$ 0 to a higher $b_0$ at SS $\sim$ 1, obviously ln(SS) must be multiplied by a higher $b_1$. I would omit the corresponding figure 6.

12) Page 12, line 19: Why? try to give a qualitative interpretation of this fact. . .

13) Page 14, line 13ff (Fig10): I would need a better motivation here, why these piecewise linear fits were tried in the first place – looking at this figure an exponential fit would be much more appropriate and at this point it is not clear what the authors are trying to do here – it gets a bit clearer later on, but a good explanation and motivation here would help. What does the linear approximation represent physically?

14) Page 14, line 16: CMDe has not been clearly defined here at first use – what is a "particle size that is used for describing the size range of each regression"?

15) Page 14, line 27: In line 22 you described the different behavior of slope and offset

here you say that they vary simultaneously? Please be precise with the language. Also "simultaneously" implies temporal variation, which is not the case here . . .

16) Page 14, line 27/28: "This is the link . . ." What precisely is the link? You discussed several variations in the sentences before, but none of them seem a clear "link" to me.

17) Page 15, line 9-13: This argument is very technical and also in my opinion the linear relationship has nothing to do with dependence on GMD and GSD (see point 12). There are much better, more qualitative arguments, why these coefficients should depend on the size distribution and how – please argue more along those lines.

18) Page 18, line 17/18: Please give an interpretation of this fact.

19) Page 19, line 15/16: What coefficients are you taking about? There are many discussed in the manuscript. Please be more precise with your language throughout the manuscript.

Minor comments:

Page 1, line 27: SAE_10 and BSF_min not defined

Page 2, line 1: supplementary -> replacement (or similar), "supplementary" means supporting or additional; please check for similar mistakes throughout the manuscript, or have it proofread by a native speaker

Page 2, line 4: Do you mean "or" instead of "of", otherwise this sentence does not make sense to me.

Page 4, line 25: What do you mean by "likewise in Smale . . . "?

Page 5, line 28; "Instrument mentors" is not a valid term, please check manuscript throughout for such wrong use of vocabulary.

Page 6, line 5: coal is also a fuel

Page 6, line 9: is this not a coastal site, rather than marine?

Page 6, line 9: "emphase" is not a word

Page 6, line 9/10: The sentence is not grammatically correct

Page 16, line 13: is -> are

---

## Author Comment (AC1) · 30 Sep 2019

acp-2019-149, Shen et al.: Estimating CCN number concentrations using aerosol optical properties: Role of particle number size distribution and parameterization

**Replies to the reviewers' comments**

The authors thank the reviewers for their evaluations. We believe answering the questions helped improving the paper. The text was corrected according to most of the suggestions of the reviewers. The largest changes were

1) In the section 1. Introduction we added text describing the geographical and temporal availability of data on aerosol optical properties, cloud condensation nuclei, and particle number size distributions. This was done as a reply to the comment of reviewer #2.

2) We added a description of two the backscatter fraction (BSF) and scattering Ångström exponent (SAE) as a new subsection 2.3 Optical properties calculated from the nephelometer data. The section shows both the equations and a qualitative description of the relationship of particle size, BSF and SAE. We also added the section 2.6 Aerosol optical properties and CCN concentrations of simulated size distributions. These additions were done in order to help discussions later in the paper.

3) In order to make the paper more clearly describing also aerosol physics, not only the parameterization, we changed the titles of the sections from

1. Introduction
2. Methods
2.1 Sites and measurements
2.2 Data processing
2.3 Light scattering calculated from the particle number size distributions
2.4 CCN number concentration calculated from the particle number size distribution
3. Overview of measured properties
3.1 AOPs and CCN calculated from particle size distributions
3.2 Relationships between AOPs and CCN
4. Development of the parameterization
4.1 Site-dependent parameterization for each measured supersaturation, $N_{CCN}(AOP_1)$
4.2 General combined parameterization $N_{CCN}(AOP_2)$
5. Results and discussion
5.1 Comparison of $N_{CCN}$ from the AOP parameterization and measurements
5.2 Evaluation of the effect of particle size distribution to the parameterization
5.3 Aerosol size characteristics for all sites
5.3.1 Diurnal variation of particle number size distribution
5.3.2 Distribution of geometric mean diameter
5.3.3 Contribution of light scattering by sub-µm particles
6. Conclusions

to

1. Introduction
2. Methods
2.1 Sites and measurements
2.2 Data processing
2.3 Optical properties calculated from the nephelometer data
2.4 Light scattering calculated from the particle number size distributions
2.5 CCN number concentration calculated from the particle number size distribution
2.6 Aerosol optical properties and CCN concentrations of simulated size distributions
3. Relationships between $N_{CCN}$ and AOPs
3.1 Site-dependent $N_{CCN}$ - AOP relationships
3.2 Site-independent relationships between $N_{CCN}$, AOPs and supersaturations
4. Analyses of size distribution effects on $N_{CCN}$ –AOP relationships
4.1 $N_{CCN}$ –AOP relationships of simulated particle size distributions
4.2 Aerosol size characteristics of the sites
4.2.1 Diurnal variation of particle number size distribution
4.2.2 Distributions of geometric mean diameters
4.3.3 Contribution of light scattering by sub-µm particles
5. Conclusions

We included more supplements in order to answer reviewers questions.

In the revised text major changes and additions are written using red letters. Minor language corrections have not been indicated.

Below the the replies are intended.

**Detailed replies to Anonymous Referee #1**

**Anonymous Referee #1**
Major and General Comments:
1) Page 1 line 26 of the Abstract is misleading. The manuscript focusses on the relationship be-tween AOD and CCN, this parametrization also depends on the s,m upersaturation, however the manuscript does not show that the relationship between measured CCN number concentration and the corresponding supersaturation (S) is logarithmic, there is even no figure and no discussion about the relationship between CCN and S in the manuscript.

> We don't quite understand the reviewer's comment because the derived formula
>
> $N_{CCN} = ((287*SAE * \ln(SS/0.093) * (BSF - constant1) + 5.2)\sigma_{sp}$
>
> includes a logarithmic dependence on SS. The section where this was derived included discussion also. However, we have now included a new figure that shows this clearly. Fig. 1 of the revised paper shows the averages of the measured $N_{CCN}$ at the six sites at the station-specific supersaturations of the CCN counters and a logaritmic fitting to the data. Also in the figures of section where the equation is derived the logarithmic and power function fittings are presented and discussed.

**2) The manuscript is very technical, following the manuscript would have fit perfect in the journal aerosol measurement techniques as well. Please clarify why this is an ACP**
**rather than an AMT manuscript.**

> Here is our argumentation for keeping the paper in ACP. In the paper we processed data from from six stations in very different environments around the world. We developed a parameterization to estimate the number concentration of cloud condensation nuclei ($N_{CCN}$) from measurements of aerosol optical properties. We have also studied how the dependence of $N_{CCN}$ on supersaturation affects the relationships of $N_{CCN}$ and AOPs. We first show that with the new Fig. 1 where NCCN vs. SS is plotted for all the stations and a logarithmic function fits the data clearly better than a Twomey-style power function. We found a formula that combines data from all six sites, it is approximately
>
> $N_{CCN} = ((286*SAE * \ln(SS/0.093) * (BSF - constant) + 5.2)\sigma_{sp}$
>
> where SAE is the scattering Ångström exponent, BSF the backscatter fraction, $\sigma_{sp}$ the total scattering coefficient and SS the supersaturation. This formula shows that $N_{CCN}$ depends on SAE, BSF and $\sigma_{sp}$ and that there is a logarithmic dependence on SS. If the data were fitted by using a power function of SS the fitting was not as good as by using a logarithm of SS. See the revised Figure 5.

[Figure]

Figure 5. The coefficients a and b of each station (Table 2) as a function of supersaturation. Two types of functions, a logarithmic and a power fuction were fitted to the coefficient a, to coefficient b only a logaritmic function. The squared correlation coefficients $R^2$ are shown only for the power function fittings, for the logarithmic fittings they were all > 0.99.

We simulated aerosol size distributions and obtained similar relationships between $N_{CCN}$ and aerosol optical properties as in the field data. The above arguments show that the formula that we present can be considered either as only a parameterization or also as a representation of a physical phenomenon, not tied to a geographical location. This can be considered atmospheric physics.

In addition we studied the uncertainty of the method and the explanations of the uncertainties based on the aerosol size distributions in different environments. This is also atmospheric physics.

These are the main arguments we considered supporting the publication in ACP.

3) Table 1 it would be very helpful to add a small description of the site.
    We have modified the table accordingly.

4) Page 6 line 14 the authors mention that no severe pollution episodes where observed in the data used for this study. However, table 1 shows that the data used for this study span a whole year. Was the data selected for no pollution periods and if so, what was the criterium?
    The data were not purposely selected, only very few (<<1%) suspicious data points were excluded. The percentage of excluded data is also added into the manuscript (supplement S1).
    We also change the paragraph accordingly.  he claim that no severe pollution episodes were observed is a qualitative, not a very precise statement.  The paragraph was replaced by
        MAO refers to Manacapuru in Amazonas, Brazil. It is a relatively clean site where Manaus pollution plumes and biomass burning plumes impact the background pristine rainforest aerosol alternately (e.g., de Sá et al., 2019).

5) Page 7 line 3-9 describes which data have been excluded for the analysis. It would be nice to mention the percentage of excluded data by the different criteriums and in total.

> Detailed description and percentages of excluded data are presented in the supplement S1.

6) Page 9 line 9-11: The authors describe that NCCN was calculated by integrating the number concentration of aerosol particles larger than a certain diameter. How these diameters were chosen? Using the Köhler Theory and a certain hygroscopicity for the particles? If yes, which hygroscopicity have been used?

> In the discussion paper we had chosen the critical diameters in the range of 80 - 110 nm intuitively for the simulation. There was no proper reasoning for this selection. When replying the reviewer's question we calculated that this corresponds to the supersaturation range of 0.213 – 0.344 when the global average hygroscopicity parameter kappa of 0.27 is used. This range is narrow so we repeated the simulations using a dritical diameter range of 50 – 150 nm which is more reasonable that corresponds to a SS range of ~0.14 – 0.74% with the same kappa.

7) Page 10 line 28: The authors mention that the error in the measurements influence the result of the linear regression. As long as the linear relationship between measured variables which both have an uncertainty plays a key role in this study, the author should use a bivariate regression including the uncertainty of all measured variables as introduced by Cantrell 2008.

> We have now repeated the calculations by using Reduced Major Axis (RMA) regression. The main results don't change although the derived function's constants are slightly different, see the supplement S2.

8) Page 11 line 18: Is there also a lower $R^2$ for higher $N_{CCN}$ concentrations? Figure 8 b suggest such a closure.

Yes, we agree with the reviewer, it shows clearly in figure 8(b) that the estimated $N_{CCN}$ is more consistent with measurements at higher CCN concentrations. We split the whole data set into two groups, one is called 'lower 50%', corresponding to the time when $N_{CCN}$(meas)<$N_{CCN}$(meass)(Median) while the rest is called 'upper 50%'. The SS for the split is set as a moderate SS% which is 0.3~0.4% depending on site. It is clear that at almost at all cases $R^2$ for 'upper 50%' groups are higher than 'lower 50%' groups. However, the subset has a narrower range of variation. The $R^2$ for 'upper 50%' groups are still lower than the $R^2$ of the whole data set. The details are shown in the table below.

| | | $R^2$ | | |
| --- | --- | --- | --- | --- |
| | | Whole | Lower 50% | Upper 50% |
| SMEAR II | 0.10% | 0.66 | 0.28 | 0.54 |
| | 0.20% | 0.84 | 0.36 | 0.65 |
| | 0.50% | 0.75 | 0.38 | 0.48 |
| | 1.00% | 0.54 | 0.38 | 0.43 |
| SORPES | 0.10% | 0.59 | 0.16 | 0.26 |
| | 0.20% | 0.75 | 0.38 | 0.73 |
| | 0.40% | 0.68 | 0.40 | 0.71 |
| | 0.80% | 0.66 | 0.33 | 0.60 |
| PGH | 0.12% | 0.81 | 0.48 | 0.73 |
| | 0.22% | 0.83 | 0.57 | 0.72 |
| | 0.48% | 0.79 | 0.63 | 0.61 |
| | 0.78% | 0.74 | 0.57 | 0.55 |
| PVC | 0.15% | 0.83 | 0.36 | 0.77 |
| | 0.25% | 0.84 | 0.35 | 0.81 |
| | 0.40% | 0.84 | 0.31 | 0.80 |
| | 1.00% | 0.81 | 0.19 | 0.78 |
| MAO | 0.25% | 0.81 | 0.32 | 0.79 |
| | 0.40% | 0.83 | 0.36 | 0.80 |
| | 0.60% | 0.74 | 0.30 | 0.65 |
| | 0.80% | 0.69 | 0.22 | 0.59 |
| ASI | 0.15% | 0.84 | 0.43 | 0.82 |
| | 0.25% | 0.74 | 0.26 | 0.64 |
| | 0.40% | 0.65 | 0.17 | 0.47 |
| | 0.80% | 0.54 | 0.18 | 0.30 |

We do not add that into the text since the paper is already long enough. However, we added this brief discussion into the paper's section 3.3:

The correlation coefficient of $N_{CCN}$(AOP$_2$) vs. $N_{CCN}$(meas) is higher at higher CCN concentrations (not shown in the figure). One possible reason for this is that when CCN concentration is lower, the aerosol loading is usually lower and also the relative uncertainties of both $N_{CCN}$ and AOPs are higher than at high concentrations.

9) Page 12 line 12: What is a reasonable long period?

Thank you for this question. We added an analysis in the supplement (S5. Analysis of the uncertainty related to the number of samples). It shows that when the number of hourly samples is > 1000 the uncertainty of the BSFmin is low enough.

10) Page 12 line 22-23: What is the percentual uncertainty?

Yes we agree it would be helpful to present the percentual uncertainty. We rewrite the paragraph as below:

Using $287 \cdot SAE_{10}$ overestimates or underestimates $a_1$ by +41%, +30%,  -20%, -32%,-26% and +252% for SMEAR2, SORPES, PGH, PVC MAO and ASI, respectively. It brings additional uncertainty in the estimation and the uncertainty may be very large is $a_1$ is very small. Nevertheless $SAE_{10}$ is the only parameter we found that positively related with $a_1$ and can easily

been obtained from measurement at current stage. Searching for a more suitable proxy for $a_1$ would be an important part of follow up studies.

11) Page 14 line 6-7: The authors wants to show the different magnitude in Figure 9a, following I would recommend to use only one axis, by that the different magnitudes would be visible easier.

In the revised version the $R_{CCN/\sigma}$ of the wide size distributions are plotted twice: the black symbols and line use the left axis to emphasize the big difference in the magnitudes of the wide and narrow size distributions; the red symbols and line use the right axis to show that the shape of the $R_{CCN/\sigma}$ size distribution is very similar to the one calculated for thew narrow size distributions.

12) Page 16 line 24: The MAO site was measuring aerosol downwind of the City Manaus and not Amazon rainforest air. The pronounced nucleation size particles suggest predominant anthropogenic emissions from Manaus.

We thank the referee for pointing this out. We modified the manuscript accordingly as below:

NPF is rarely observed in the Amazon forest as reported by Wang et al. (2016). However, it does take place at MAO as is shown in the diurnal cycle of PNSD. The reason is probably that the MAO site was measuring aerosol downwind of the City Manaus. At ASI, there no evidence of NPF according to the PNSD diurnal cycle.

13) Page 17 line 14: The authors compares the result from MAO with the results from ATTO. The disagreement of this comparison is a strong indication that the air masses measured at MAO are not representative for the Amazon rainforest. Please clarify this aspect.

We thank the referee for pointing this out. It was a mistake to make a comparison between ATTO and MAO. Now we clearly realize that even though ATTO and MAO are only ~100km from each other they represent different location and air masses. We removed the comparison from the manuscript.

Minor Comments
- The authors always write 'SS%' instead of 'SS' or 'S' for the supersaturation. The '%' behind the value is sometimes used, sometimes not. I would recommend to use 'SS' or 'S' for the supersaturation and always mention the unit behind a variable.

We have changed all SS% symbols to SS according to the reviewer's suggestion

- Page 13 line 5: two times "of"
corrected

- Page 14 line 9: is this a complete sentence?
No. The sentences " Note also that the rates of decrease of $R_{CCN/\sigma}$ and BSF. We used this information for estimating particle sizes with a stepwise linear regression." were replaced by
"The decrease of $R_{CCN/\sigma}$ and BSF with growing GMD was used for estimating particle sizes with a stepwise linear regression."

- Page 16 line 1: is the double negation on purpose?
No, it was a typing error. Corrected.

**Anonymous Referee #2**

The paper is on an interesting subject and the parameterization of CCN concentrations by optical parameters can be useful. The data are carefully analyzed and the parameterization is developed step by step, which helps understanding the resulting complicated formula. Overall, I find that this is a paper that is in principle publishable after moderate to major revisions. One reservation I have, is that it is clearly a method paper that can certainly be very useful, but does not have major scientific results (as to e.g., actual CCN in various environments and larger implication), so ACP is in my opinion not the best journal for this. It fits much better in a method-oriented journal. I would strongly suggest to submit it somewhere else.

> Reviewer 1 asked the same question. See the reply above.

Parts of the steps taken could be better motivated and better interpreted (see comments below), which would make the manuscript easier to follow and less technical and dry. Overall I think the length of the paper and the 13! Figures are a bit excessive. Please think of a way to summarize the information in a more compact way.

> We have revised the text to make it a bit less technical. This was done by changing the titles of the sections to more descriptive ones and by adding more interpretations related to size distributions at the different sites. We removed Fig 3. of the discussion paper but in order to answer reviewer #1's question 1) about showing the logarithmic dependence of $N_{CCN}$ on SS we added a new Fig 1. The number of figures is still 13. There are several papers also in ACP that have 13 figures so we don't consider that to be too many.

The writing needs to be improved overall and the English corrected in some places (many formulations are unclear and sometimes grammatically incorrect).

> We have now corrected many parts of the paper.

Specific comments:
1) Page 3, line 4: Explain better, why they are not adequate. I assume you mean the spatial coverage? Are optical properties really more widely measured than size distributions? Support this claim!

> Yes, the spatial coverage of particle size distribution data is indeed clearly smaller than that of the optical properties. The WMO Global Atmosphere Watch World Data Centre for Aerosols (GAW-WDCA) (http://ebas.nilu.no/) contains on 20 June 2019 particle number size distribution data sets from 22 countries altogether from 58 stations, but only 5 are outside of Europe. The CCNC data are from 3 European sites. Light scattering coefficients measured with a nephelometer are from 31 countries and 103 stations that are located on all continents and also on some islands. The temporal coverage data in the GAW-WDCA data base is such that the oldest nephelometer data, those from Mauna Loa start in 1974 whereas the oldest particle number size distribution data, those from SMEAR II start in 1993. Another easily available source for data is the US Department of Energy Atmospheric Radiation Measurement (ARM) user facility (https://www.arm.gov/data). On 20 June 2019 we found that the ARM research facility data contains particle size distribution data from 7 permanent sites and light scattering coefficient measured with a nephelometer at 20 sites. It is clear that there are other data sets of all of these measured around the world but those that can be found either from the GAW-WDCA or the ARM data bases are are quality controlled and readily availabe.

2) Page 3, line 26: Do you mean N_CCN/sigma_sp instead of R_CCN/sigma_sp?
Later you call this ratio R_(CCN/sigma) please be consistent.

> This is a good observation, we have corrected this.

3) Page 8, line 6-9: This paragraph consists of several incomplete, grammatically incorrect sentences and as a result, it is unclear what you are actually doing in this calculation. Please formulate clearly and explain in more detail.

        Corrected.

4) Page 9, line 5: How well do you think a unimodal size distribution can describe a realistic situation, where size distributions are at least bi-modal? Give some motivation already here for choosing this approach.

        The following paragraph was added:

Using a unimodal size distribution for the simulation is an approximation only. In the boundary layer particle number size distributions consist typically of an Aitken mode in the size range ~25 – 100 nm, an accumulation mode in the size range 100 – 500 nm, and during new particle formation also a nucleation mode in the size range < 25 nm in very different environments (e.g., Dal Maso et al., 2005; Herrmann et al., 2015; Qi et al., 2015). While the total number concentration is dominated by the smaller modes, essentially all light scattering is due to the accumulation mode and also coarse particles in the range of 1 - 10 µm. For instance at SMEAR II the average contribution of particles smaller than 100 nm to total scattering was ~0.2 % and even at the end of new particle formation events not more than ~2% (Virkkula et al., 2011). On the other hand, also most of the CCN are also in accumulation mode range, especially for low supersaturations (SS < 0.2%), at higher SS also in the Aitken mode (Sihto et al., 2011).

5) Page 9, line 13: The section is called "Overview of measured properties", but it does not contain an overview of measured properties, more a quality control. It should be merged with the method section.

        We agree, as it is in the discussion paper the section really is not an overview of the data. But now we have rewritten the beginning of section 3. First, the titles were changed from

                3. Overview of measured properties
                3.1 AOPs and CCN calculated from particle size distributions
                3.2 Relationships between AOPs and CCN
    to

        3. Relationships between $N_{CCN}$ and AOPs
        3.1 Site-dependent $N_{CCN}$ - AOP relationships
        3.2 Site-independent relationships between $N_{CCN}$, AOPs and supersaturations

        Second, an overview of the data at the different sites is given in the new section 3.1.

6) Page 9, line 14-24: This Paragraph seems out of place here. At the beginning of a new section, before the sub-sections, I would expect some general introduction.Instead we get some details of only one station that are neither well explained nor well motivated. Please rearrange this section and find a better place for this information (after you clarify it).

        We agree and have modified the paragraph  and the section 3. We have added a short introduction at the beginning of the sections.

7) Page 9, line 19: You do not make clear how you make this fit (e.g., how you determine k) and why it does not fit. Is this relevant for the paper in any way? It well known that an error function gives a much better fit to CCN spectra.

        The reviewer is right, this is not relevant. We removed this. We also rewrote the beginning of section 3.

8) Page 9, line 24: From the previous discussion it is not clear, why/how this parameterization involved absorption data. If this paragraph is absolutely necessary, please explain in more detail, otherwise I would suggest to omit this discussion entirely.

        The reviewer is right, this is not relevant. We removed this.

9) Page 10, line 13/14: Give an interpretation of this fact and a motivation, why you investigate this dependence.

The intepretation is that when BSF is constant the shape of the particle size distribution is also very probably constant in the particle size range significantly affecting light scattering. Then the increase of scattering coefficient is very probaly due to the increase in the particle number concentration in the same size range and then also the number of CCN. Fig. 1 alone does not show how NCCN depends on BSF so it serves as a motivation of the continuation to the use of Eq. (3) and the subsequent analyses.

10) Page 11, line 11/12: Please use the appropriate reduced major axis regression then, instead of a regression that is not suitable for your data

We have now repeated the calculations by using Reduced Major Axis (RMA) regression. The main results don't change although the derived function's constants are slightly different, see the supplement S2.

11) Page 12, line 3/4: I am not sure what you mean by "underlying reason"? b0 is the limiting value for SS=1 and the range of SS stays the same. So go from Nccn 0 at approx. SS 0 to a higher b0 at SS 1, obviously ln(SS) must be multiplied by a higher b1. I would omit the corresponding figure 6.

The scatter plot in Fig. 6 is probably the most important one in the paper since it shows that the constants are linearly dependent on each other. The linear relationships are then used to reduce free parameters. The regressions yield the constants needed for the development of the general parameterization, Eq. (6). We do not omit Fig. 6.

It is true that $b_0$ is the limiting value for SS = 0% but the absolute values of the constants are by no means obtained from the data without the regressions. First, not all the stations even have measurements at SS = 1% so there is no way to deduce the values of these constants without the fitting to the data. Second, it is by no means clear that the constants of all the stations would follow straight lines. For instance, why would the pairs ($b_1,a_1$) of all the stations be distributed approximately along straight lines? But plotting them to the same figure they do. Why wouldn't the forms of the scatter plots be nonlinear? Or even random? There is no a priori reason. This is what we try to study by the simulations in section 5.2. And that is what we we wondering by writing about an "underlying reason". But all right, we remove this expression.

12) Page 12, line 19: Why? try to give a qualitative interpretation of this fact. . .

The coefficient $a_1$ is positively correlated with SAE which means that the higher SAE is the higher is $N_{CCN}$ at a constant $\sigma_{sp}$. Light scattering and its wavelength dependency is highly dependent on particle size so SAE is often used as an indicator of the particle size distribution. It is often inversely related to particle size: at SAE > 2, the volume size distribution is typically dominated by particles smaller than 0.5 µm, and at SAE < 1, larger particles ($D_p$ > 0.5 µm) dominate the distribution. So, the positive correlation of SAE and $a_1$ means that when volume size distribution is dominated by small particles the CCN concentration is high. We have added several qualitative interpretations into the text.

13) Page 14, line 13ff (Fig10): I would need a better motivation here, why these piecewise linear fits were tried in the first place – looking at this figure an exponential fit would be much more appropriate and at this point it is not clear what the authors are trying to do here – it gets a bit clearer later on, but a good explanation and motivation here would help. What does the linear approximation represent physically?

It is true that an exponential fit would be perfect to explain the relationship between $R_{CCN/\sigma}$ and BSF. But this is not at all what we are looking for. We are looking for the slopes and offsets in the relationship

$R_{CCN/\sigma}$ = aBSF + b that was used for fitting the field measurement data. So, physically it would mean that $N_{CCN}$ would increase linearly as a function of BSF even though this is not exactly correct.

We assume that the simulated data consists of a series of lognormal size distributions. This would be simulating some station data where the GMD of size distributions varies in a certain range. Then application of linear regression yields the slopes and offsets and we can find out how these are related to each other. As it proves out, the simulated b vs a pairs follow the lines that are similar to those obtained by plotting the slopes and offsets of the actual field data (Fig. 12).

14) Page 14, line 16: CMDe has not been clearly defined here at first use – what is a "particle size that is used for describing the size range of each regression"?

On lines 16 – 18 it is written " ... *we define here as the equivalent geometric mean diameter GMD$_e$, the geometric mean of the range of the GMDs of the unimodal size distributions used for each regression.*" Geometric mean of the range of the GMDs is the square root of the product of smallest GMD and the largest GMD of the range. We do not understand how this is could be more clearly defined. It can also be presented with symbols as $GMD_e = \sqrt{GMD_{low}GMD_{high}}$ , where $GMD_{low}$ and $GMD_{high}$ are the smallest GMD and the largest GMD of the range, respectively. But it is probably clarified with examples. Above, on line 12 two examples were given of the regressions, one calculated for the GMD range from 50 nm to 100 nm and the other for the GMD range from 100 nm to 200 nm. The GMD$_e$s of these two size ranges are 70.7 nm and 141.4 nm, respectively.

15) Page 14, line 27: In line 22 you described the different behavior of slope and offset here you say that they vary simultaneously? Please be precise with the language. Also "simultaneously" implies temporal variation, which is not the case here . . .

We removed the sentence "However, they decrease and increase simultaneously." and modified the corresponding paragraph. It was replaced by

Note also that the ranges of the absolute values of the slopes and offsets of the narrow and wide size distributions are very different. For instance, when $GMD_e = 100$ nm the slope a $\approx 4000$ cm$^{-3}$/Mm$^{-1}$ and a $\approx 1600$ cm$^{-3}$/Mm$^{-1}$ for the narrow and wide size distribution, respectively. Since $N_{CCN} = R_{CCN/\sigma} \cdot \sigma_{sp} = (aBSF + b)\sigma_{sp}$ this means that the $N_{CCN}$ of narrow size distributions is more sensitive to variations in mean particle size than the $N_{CCN}$ of wide size distributions.

16) Page 14, line 27/28: "This is the link . . ." What precisely is the link? You discussed several variations in the sentences before, but none of them seem a clear "link" to me.

We removed the sentence " This is the link to the observations from the field stations." and rewrote the corresponding paragraph. See the answer to the referee comment 15 above.

17) Page 15, line 9-13: This argument is very technical and also in my opinion the linear relationship has nothing to do with dependence on GMD and GSD (see point 12). There are much better, more qualitative arguments, why these coefficients should depend on the size distribution and how – please argue more along those lines.

The reviewer is right. Our argument was wrong, the simulations did not show the linear relationships between the coefficients $a_0$, $a_1$, $b_0$ and $b_1$ that are the factors from the fittings of $(a_1\ln(SS)+a_0)BSF + b_1\ln(SS)+ b_0$ with the data. The simulations yielded only the slope a and the offset b of aBSF + b but not their dependence on SS. To obtain the factors $a_0$, $a_1$, $b_0$ and $b_1$ we should have added also a simulation as a function of SS. That was not done. In order not to add new figures, we did not include a figure where the simulated a vs SS and b vs SS is shown.

However, the simulation clearly shows that both the slope a and the offset b depend very strongly on both the geometric mean diameter (GMD) and the geometric standard deviation (GSD) of the size distribution (Fig. 11). We do not see any other way but the simulation to show the dependence of a and b on the size distribution.

We have replaced the corresponding paragraph by

Several observations can be made of Fig. 11. First, for the simulated wide size distributions the relationship of the offset and slope is unambiguous but not for the narrow size distributions at sizes $GMD_e > \sim 200$ nm (Fig 11b). Secondly, the field data points roughly follow the lines of the simulations. This suggests that the slopes and offsets of the linear regressions of $R_{CCN/\sigma}$ vs. BSF yield information on the dominating particle sizes just as they do for the simulated size distributions. For instance, the PVC data point corresponding to the highest supersaturation has the highest slope (1970 cm$^{-3}$/Mm$^{-3}$, Table 3) and it is close to the wide size distribution line with the activation diameter of 50 nm (Fig. 11a). This corresponds to the $GMD_e$ of $\sim 150$ nm (Fig. 11b). The SMEAR II high SS offset vs. slope fits best with the corresponding lines of the narrow unimodal size distributions with activation diameters in the range of $\sim 50 - 110$ nm and the corresponding $GMD_e \approx 150 - 200$ nm.

18) Page 18, line 17/18: Please give an interpretation of this fact.
    The reviewer refers to this sentence: " However, a comparison of the correlation coefficients and the $\sigma_{sp}(PM_1)/\sigma_{sp}(PM_{10})$ frequency distributions of each site shows the opposite. " and to the list of sites in an order of decreasing correlation coefficients in the next sentence.
    This means that the correlation coefficient of $N_{CCN}$ calculated from the aerosol optical properties ($N_{CCN}(AOP_2)$) vs. the $N_{CCN}$ measured directly ($N_{CCN}$(meas)) is the higher the larger fraction of total scattering is due to large, $PM_{10}$ aerosol. This suggests that $N_{CCN}$ can be estimated better from the aerosol optical properties for sites dominated by large particles than for sites dominated by small particles. This further suggests that the ambient size distributions were so wide that the nonmonotonous relationship between particle size and  BSF discussed above did not play an important role.

19) Page 19, line 15/16: What coefficients are you taking about? There are many discussed in the manuscript. Please be more precise with your language throughout the manuscript.
    The sentence is about the coefficients $a_0$, $a_1$, $b_0$ and $b_1$ in the formula
    $N_{CCN}(AOP_2) = (aBSF + b)\sigma_{sp} = (a_1\ln(SS)+a_0)BSF + b_1\ln(SS)+ b_0)\sigma_{sp}$.

Minor comments:
Page 1, line 27: SAE_10 and BSF_min not defined
    We replaced $SAE_{10}$ simply by SAE since it was just in the previous sentence written that the parameterization deals with PM10 particles. BSFmin has now been defined.

Page 2, line 1: supplementary -> replacement (or similar), "supplementary" means supporting or additional; please check for similar mistakes throughout the manuscript, or have it proofread by a native speaker
    The sentence
        This makes particle number size distribution measurements capable of serving as a supplementary of direct CCN measurements.
    was replaced by
        This makes particle number size distribution measurements capable of substituting direct CCN measurements.

Page 2, line 4: Do you mean "or" instead of "of", otherwise this sentence does not make sense to me.

Yes, "of" was a typing error. Corrected.

Page 4, line 25: What do you mean by "likewise in Smale . . . "?

"likewise in" was replaced by "similar to"

Page 5, line 28; "Instrument mentors" is not a valid term, please check manuscript throughout for such wrong use of vocabulary.

"Instrument mentors" were removed and the sentence

The supersaturations are typically calibrated before and after each campaign at an altitude similar to measurement site by instrument mentors according to CCN handbook (Uin, 2016b).

was replaced by

The supersaturations are typically calibrated before and after each campaign at an altitude similar to measurement site according to the CCN handbook (Uin, 2016b).

Page 6, line 5: coal is also a fuel

" coal and fuel combustion " was replaced by " coal combustion "

Page 6, line 9: is this not a coastal site, rather than marine?

Yes, Cape Cod is a coastal site, not marine. Replaced.

Page 6, line 9: "emphase" is not a word

The sentence where the word "emhase" was is

"Ascension Island (ASI) locates in the southeast Atlantic where westward transport of southern Africa biomass-burning aerosols emphases heavy aerosol loading."

It was replaced by

"Ascension Island (ASI) is located in the southeast Atlantic where westward transport of biomass-burning aerosols from southern Africa may increase aerosol concentrations to high levels."

Page 6, line 9/10: The sentence is not grammatically correct

The paragraph on ASI was corrected.

Page 16, line 13: is -> are

Corrected

---

## Author Comment (AC3) · 30 Sep 2019

**Supplement**

The Supplement contains the following sections:

**S1. Amount of data used for the analyses, fractions of accepted data and criteria for data removal**

Suspicious data within the whole dataset were removed according to the following criteria:

1)   For the size distribution data, all the data with unexplainable spikes were removed manually;

There were 7587 available hourly-averaged-PNSD in MAO with 104 bins of each. A total of 5234 spikes were removed. This accounts for ~0.7% of the total number of bins. 423 out of 7587 (~5.6%) distributions had at least 1 bin(s) removed. A distribution with few missing bins are still usable if treated properly. Only 55 (~0.7%) distributions had more than 10 spikes removed.

Besides for MAO, other data sets rarely suffered from such spikes. 32 out of 11502 (~0.3%) distributions were removed for ASI.   For SORPES and SMEAR2, less than 1% of distributions were removed. We didn't remove anything from PNSD of PVC and PNSD is not available in PGH.

2) for CCN measurements, insufficient water supply may cause underestimation of CCN, especially at lower supersaturation ratios (DMT, 2009). $N_{CCN}$ reading at lower SS% has a sudden drop a few hours before the similar sudden drop for higher SS% under such conditions, so data from such periods were removed;

Besides from the QC flag within MAO dataset, additional 55, 112,120 and 123 data points were removed at SS=0.25%,0.4%, 0.6% and 0.8% respectively, which accounts for ~0.7%-1.6% of total available data. For SORPES and SMEAR2 ~1% of total available data were removed. For ASI, PVC and PGH, no further treatment was applied besides the original QC flag.

3) if any obvious inconsistencies between the AOPs and PNSD or between the $N_{CCN}$ and PNSD were found on closure study, all the data in the same hour were removed.

51 successive hours of data from PVC were removed before analysis, which account for ~3% of the data we used in this study. 84 sparse data points were removed from the ASI data set, which account for ~0.7% of total available data. For SORPES and SMEAR2 less than 1% of data were removed.

In total, additional quality control removes ~2%, ~3%, ~1% and 0% of the total available data in MAO, PVC, ASI and PGH respectively. The exact number for SORPES and SMEAR2 is not applicable since those 3 criteria are within the original data process procedures. However, a rough estimation of fractional data removed by such criteria are 0.5%~2%.

The total number of available hourly-averaged data, accepted data and removed data and the fractions of these are presented in Table TS1.

Table TS1. Number of data and fractions of removed data from all stations

| | Period (hours) | | AOPs | | | CCN | | | Size distribution | | |
|---|---|---|---|---|---|---|---|---|---|---|---|
| | | | from Dataset | Addtional QC | finalized data | from Dataset | Addtional QC | finalized data | from Dataset | Addtional QC | finalized data |
| SMEAR2 | 8784 | N_total | -- | -- | 8626 | -- | -- | 6973-6994 | -- | -- | 8461 |
| | | Percentage | -- | -- | 98.2% | -- | -- | 79.3-79.6% | -- | -- | 96.3% |
| SORPES | 8760 | N_total | -- | -- | 5266 | -- | -- | 4825~4906 | -- | -- | 5440 |
| | | Percentage | -- | -- | 60.1% | -- | -- | 55.1~56% | -- | -- | 62.1% |
| ASI | 12144 | N_total | 11851 | 84 | 11767 | 9894-10343 | -- | 9894-10343 | 10931 | 32 | 10899 |
| | | Percentage | 97.6% | 0.7% | 96.9% | 81.5-85.2% | -- | 81.5-85.2% | 90.0% | 0.3% | 89.7% |
| PVC | 1800 | N_total | 1637 | -- | 1637 | 1495 | -- | 1495 | 1730 | 0 | 1730 |
| | | Percentage | 90.9% | -- | 90.9% | 83.1% | -- | 83.1% | 96.1% | 0.0% | 96.1% |
| MAO | 8160 | N_total | 7532 | -- | 7532 | 7574-7653 | 55-123 | 7507-7541 | 7587 | 56 | 7541 |
| | | Percentage | 92.3% | -- | 92.3% | 92.8~93.8% | 0.7-1.5% | 92-92.4% | 93.0% | 0.7% | 92.4% |
| PGH | 3498 | N_total | 3453 | -- | 3453 | 3380-3420 | -- | 3380-3420 | -- | -- | -- |
| | | Percentage | 98.7% | -- | 98.7% | 96.6-97.8% | -- | 96.6-97.8% | -- | -- | -- |

**S2. Application of Reduced Major Axis (RMA) regression**

The Matlab code of Trujillo-Ortiz and Hernandez-Walls (2010) was applied to calculate the reduced major axis (RMA) regressions of $R_{CCN/\sigma}$ vs. BSF to get the slope and offset (a and b, respectively) of $R_{CCN/\sigma} = a\,\text{BSF} + b$ at the supersaturations (SS) of the CCN counters at the six stations. The results are shown in Table TS2. The values of Table TS2 were plotted as a function of SS in Fig. SF1 where also the fittings to the data are shown.

Table TS2. Slopes (a) and offsets (b) of $R_{CCN/\sigma} = a\,\text{BSF} + b$ obtained with RMA.
The unit of the coefficients is $[N_{CCN}]/[\sigma_{sp}] = \text{cm}^{-3}/\text{Mm}^{-1}$.

| Station | SS(%) | a | ($a_{LOW}$ - $a_{HIGH}$) | b | ($b_{LOW}$ - $b_{HIGH}$) |
|---|---|---|---|---|---|
| SMEAR II | 0.1 | 175 | ( 170 - 181) | -15.0 | ( -15.8 - -14.3) |
| | 0.2 | 511 | ( 502 - 521) | -49.8 | (-51.2 - -48.5) |
| | 0.5 | 1031 | ( 1011 - 1050) | -110.1 | (-112.9 - -107.3) |
| | 1 | 1492 | ( 1459 - 1525) | -164.4 | (-169.1 - -159.7) |
| SORPES | 0.1 | 121 | ( 117 - 125) | -9.1 | ( -9.5 - -8.7) |
| | 0.2 | 333 | ( 326 - 341) | -25.8 | ( -26.6 - -25.0) |
| | 0.4 | 657 | ( 643 - 671) | -53.0 | ( -54.6 - -51.5) |
| | 0.8 | 926 | ( 905 - 946) | -76.6 | ( -78.9 - -74.4) |
| PGH | 0.12 | -53 | ( -54.6 - -51) | 5.1 | ( 5.0 - 5.2) |
| | 0.22 | 161 | ( 156 - 167) | -6.9 | ( -7.3 - -6.5) |
| | 0.48 | 712 | ( 689 - 734) | -37.6 | ( -39.2 - -36.0) |
| | 0.78 | 849 | ( 823 - 876) | -44.1 | ( -46.0 - -42.3) |
| PVC | 0.15 | 517 | ( 500 - 534) | -42.4 | ( -44.5 - -40.3) |
| | 0.25 | 989 | ( 956 - 1023) | -85.8 | ( -89.9 - -81.7) |
| | 0.4 | 1465 | ( 1416 - 1514) | -130.7 | ( -136.7 - -124.7) |
| | 1 | 2452 | ( 2369 - 2536) | -223.5 | ( -233.7 - -213.3) |
| MAO | 0.25 | 472 | ( 462 - 481) | -46.7 | ( -48.1 - -45.4) |
| | 0.4 | 833 | ( 817 - 849) | -83.4 | ( -85.6 - -81.1) |
| | 0.6 | 1188 | ( 1163 - 1213) | -122.1 | ( -125.6 - -118.7) |
| | 1.1 | 2128 | ( 2065 - 2190) | -226.5 | ( -234.9 - -218.2) |
| ASI | 0.1 | 150 | ( 147 - 153) | -15.9 | ( -16.3 - -15.4) |
| | 0.2 | 319 | ( 312 - 325) | -34.0 | ( -34.9 - -33.1) |
| | 0.4 | 372 | ( 365 - 380) | -39.8 | ( -40.9 - -38.7) |
| | 0.8 | 406 | ( 397 - 414) | -42.4 | ( -43.6 - -41.1) |

[Figure]

Figure SF1. The RMA-derived coefficients a and b of each station (Table ST2) as a function of supersaturation. Two types of functions, a logarithmic and a power fuction were fitted to the coefficient a, to coefficient b only a logaritmic function. The squared correlation coefficients $R^2$ are shown only for the power function fittings, for the logarithmic fittings they were all > 0.99. The unit of the coefficients is $[N_{CCN}]/[\sigma_{sp}] = cm^{-3}/Mm^{-1}$.

[Figure]

Figure SF2. Relationship between the coefficients $a_0$, $a_1$, $b_0$ and $b_1$ shown in Fig. SF1 that were obtained from the fitting of $a = a_1ln(SS) + a_0$ and $b = b_1ln(SS) + b_0$ with the data in Table TS2. SF1. a) $a_0$ vs. $a_1$, b) $b_0$ vs. $b_1$, c) $b_1$ vs. $a_1$, d) $b_0$ vs. $a_0$. The unit of the coefficients is $[N_{CCN}]/[\sigma_{sp}] = cm^{-3}/Mm^{-1}$.

1 When using RMA-derived slopes and offsets of $R_{CCN/\sigma} = a$ BSF + $b$ the relationship between
2 the factor $a_1$ and SAE became $a_1 \approx 391 \cdot SAE_{10}$ (Fig. SF3).

[Figure]

5 Figure SF3: Relationship between RMA-derived $a_1$ and $SAE_{10,median}$.
6
7 This was further used to estimate CCN number concentration in the formula

$$N_{CCN}(RMA) \approx \left( \ln\left( \frac{SS}{0.12 \pm 0.02} \right) a_1 (BSF - BSF_{min}) + R_{min} \right) \sigma_{sp} \qquad \text{(ES1)}$$

9 The derivation of (ES1) is presented in supplement S4. $N_{CCN}(RMA)$ is in general in agreement
10 with the $N_{CCN}(AOP_2)$ and $N_{CCN}(meas)$. However for SS~0.1% the performance of RMA method
11 is poor. At SS~0.1%, $R^2$ between $N_{CCN}(RMA)$ and $N_{CCN}(meas)$ is much lower than between
12 $N_{CCN}(AOP_2)$ and $N_{CCN}(meas)$ which indicates using RMA gives very uncertain results ast
13 lowest SS. Nevertheless, for SS>0.15%, OLS-derived $N_{CCN}(AOP_2)$ and RMA-derived
14 $N_{CCN}(RMA)$ agree well. Figure SF4 shows the scatter plots for $N_{CCN}(RMA)$ vs. $N_{CCN}(meas)$ and
15 $R^2$ and bias. The $R^2$ are between 0.5~0.85 and bias are within 0.5~2 when SS>0.15% for
16 $N_{CCN}(AOP_2)$.
17
18
19
20

[Figure]

Figure SF4. Statistics of $N_{CCN}$(RMA) from parameterization in Eq. (ES1). $N_{CCN}$(RMA) vs. $N_{CCN}$(meas) at different sites at different supersaturations, bias = $N_{CCN}$(RMA)/ $N_{CCN}$ (meas) at different sites and supersaturations, and $R^2$ of the linear regression of $N_{CCN}$(RMA) vs. $N_{CCN}$ (meas) at different sites and supersaturations. same as Figure 8, but for $N_{CCN}$(RMA).

**The choice between OLS and RMA**

Many studies use the reduced major axis (RMA) method instead of ordinary least squares (OLS) method to define a line of best fit for a bivariate relationship when variable represented on the X-axis contains measurement error. Smith (2009) point out that the major difference RMA and OLS is not in the difference in the assumption made about the distribution of error, but in their symmetry/asymmetry property. The reduced major axis regression is to describe the symmetric relationship between two variables and not for predictive use of the variable x with respect to y or y with respect to x (Smith, 2009). For predictive use OLS is preferred.

**References**

Smith, R. J.: Use and Misuse of the Reduced Major Axis for Line-Fitting, Am. J. Phys. Anthropol., 140, 476–486, doi:10.1002/ajpa.21090, 2009

Trujillo-Ortiz, A. and Hernandez-Walls, R.: gmregress: Geometric Mean Regression (Reduced Major Axis Regression), a MATLAB file available at: http://www.mathworks.com/matlabcentral/fileexchange/27918-gmregress, 2010.

**S3. Derivation of Equation (8)**

**a) Using slopes and offsets from ordinary linear regressions**

$$N_{CCN}(AOP) = \left(a_{ss}BSF + b_{ss}\right)\sigma_{sp} = \left(\left(a_1\ln(SS)+a_0\right)BSF + b_1\ln(SS)+b_0\right)\sigma_{sp}$$

$$R_{CCN/\sigma} = \frac{N_{CCN}(AOP)}{\sigma_{sp}} = a_{ss}BSF + b_{ss} = \left(a_1\ln(SS)+a_0\right)BSF + b_1\ln(SS)+b_0$$

Linear regressions of the coefficients in Table 2 yield

$$a_0 \approx (2.38\pm0.06)a_1, b_0 \approx (2.33\pm0.03)b_1, b_1 \approx -(0.096\pm0.013)a_1 + (6.0\pm5.9)$$

$$\Rightarrow$$

$$a_1\ln(SS)+a_0 \approx a_1\ln(SS)+(2.38\pm0.06)a_1 \approx a_1(\ln(SS)+(2.38\pm0.06))$$

$$b_1\ln(SS)+b_0 \approx b_1\ln(SS)+(2.33\pm0.03)b_1 = b_1(\ln(SS)+(2.33\pm0.03))$$

$$\approx \left(-(0.096\pm0.013)a_1 + (6.0\pm5.9)\right)(\ln(SS)+(2.33\pm0.04))$$

$$\Rightarrow$$

$$R_{CCN/\sigma} = \left(a_1\ln(SS)+a_0\right)BSF + b_1\ln(SS)+b_0$$

$$\approx a_1(\ln(SS)+(2.38\pm0.06))BSF + \left(-(0.096\pm0.013)a_1 + (6.0\pm5.9)\right)(\ln(SS)+(2.33\pm0.03))$$

Approximation, since (2.33±0.03) ≈ (2.38±0.06)

$$\Rightarrow$$

$$R_{CCN/\sigma} \approx a_1\left(\ln(SS)+(2.38\pm0.06)\right)BSF - (0.096\pm0.013)a_1\left(\ln(SS)+(2.38\pm0.07)\right) + (6.0\pm5.09)\left(\ln(SS)+(2.38\pm0.06)\right)$$

$$\approx a_1\left(\ln(SS)+(2.38\pm0.06)\right)(BSF-(0.096\pm0.013)) + (6.0\pm5.9)(\ln(SS)+(2.38\pm0.06))$$

$$\approx \left(\ln(SS)+(2.38\pm0.06)\right)\left(a_1(BSF-(0.096\pm0.013)) + (6.0\pm5.9)\right)$$

$$\approx \left(\ln(SS)-\ln(0.093\pm0.006)\right)\left(a_1(BSF-(0.097\pm0.013)) + (6.0\pm5.9)\right)$$

$$\approx \ln\left(\frac{SS}{0.093\pm0.006}\right)\left(a_1(BSF-(0.096\pm0.013)) + (6.0\pm5.9)\right)$$

**b) Using slopes and offsets from reduced major axis regressions**

$$N_{CCN}(RMA) = \left(a_{ss}BSF + b_{ss}\right)\sigma_{sp} = \left(\left(a_1\ln(SS)+a_0\right)BSF + b_1\ln(SS)+b_0\right)\sigma_{sp}$$

$$R_{CCN/\sigma} = \frac{N_{CCN}(RMA)}{\sigma_{sp}} = a_{ss}BSF + b_{ss} = \left(a_1\ln(SS)+a_0\right)BSF + b_1\ln(SS)+b_0$$

RMA regressions $\Rightarrow a_0 \approx (2.11\pm0.16)a_1, b_0 \approx (2.02\pm0.16)b_1, b_1 \approx -(0.108\pm0.016)a_1 + (7.7\pm11.0)$

The same steps as above in (a) $\Rightarrow$

$$R_{CCN/\sigma} \approx \left(\ln(SS)+(2.11\pm0.16)\right)\left(a_1(BSF-(0.108\pm0.016)) + (7.7\pm11.0)\right)$$

$$\approx \left(\ln(SS)-\ln(0.12\pm0.02)\right)\left(a_1(BSF-(0.108\pm0.016)) + (7.7\pm11.0)\right)$$

$$\approx \ln\left(\frac{SS}{0.12\pm0.02}\right)\left(a_1(BSF-(0.11\pm0.02)) + (8\pm11)\right)$$

1 **S4. Derivation of Equation (9)**

2 **If the original slopes and offsets were calculated using ordinary linear regressions**

$$N_{CCN}(AOP) \approx \ln\left(\frac{SS}{0.093 \pm 0.006}\right)\left(a_1(BSF - BSF_{min}) + C\right)\sigma_{sp},$$

where C is an unknown constant.

If $BSF = BSF_{min}$

$$\Rightarrow a_1(BSF - BSF_{min}) = 0$$

$$\Rightarrow N_{CCN}(AOP, BSF_{min}) \approx \ln\left(\frac{SS}{0.093 \pm 0.006}\right) C \cdot \sigma_{sp}$$

$$\Leftrightarrow C \approx \frac{1}{\ln\left(\frac{SS}{0.093 \pm 0.006}\right)} \frac{N_{CCN}(AOP, BSF_{min})}{\sigma_{sp}} \approx \frac{1}{\ln\left(\frac{SS}{0.093 \pm 0.006}\right)} R_{min}$$

$$\Rightarrow$$

$$N_{CCN}(AOP) \approx \ln\left(\frac{SS}{0.093 \pm 0.006}\right)\left(a_1(BSF - BSF_{min}) + \frac{1}{\ln\left(\frac{SS}{0.093 \pm 0.006}\right)} R_{min}\right)\sigma_{sp}$$

3 $$\approx \left(\ln\left(\frac{SS}{0.093 \pm 0.006}\right)a_1(BSF - BSF_{min}) + R_{min}\right)\sigma_{sp}$$

4 **If the original slopes and offsets were calculated using reduced major axis regressions**

5 $$N_{CCN}(RMA) \approx \left(\ln\left(\frac{SS}{0.12 \pm 0.02}\right)a_1(BSF - BSF_{min}) + R_{min}\right)\sigma_{sp}$$

**S5. Analysis of the uncertainty related to the number of samples**

The following procedure was used for testing how different values would be change if the number of samples decrease.

1.  For each site 2%,3%,5%,10%,20%,30%,50% and 100% of samples were taken from the whole period.
2.  The slope and offset a, b, $BSF_{min}$ (calculated as the 1st percentile of the BSF data) and $SAE_{10}$,median were calculated from the randomly chose subsets.
3.  The a, b, $BSF_{min}$ and $SAE_{10}$,median should be slightly different if the sub-set is different. Therefore the random sampling was repeated 100 times resulting in 100 different results
4.  The averages and standard deviations of the 100 results were calculated and plotted below for all the sites. The average are the reds circles and the stds the error bars in the plots.

**Results of the analysis**

The averages of a,b, $BSF_{min}$ and $SAE_{10}$,median don't have clear dependence on the number of samples. However, the uncertainty is very large at low number of samples and decreases with increasing number of samples. The uncertainties depend on parameter and site. The plots suggest that if the number of samples is larger than 1000 the uncertainty is low enough. For example, the std of $BSF_{min}$ is ~0.0005-0.005 and the std of $SAE_{10}$,median is ~0.01-0.02. For a and b, std is ~10% of the a average value.

[Figure]

Figure SF5. A monte-carlo test on the depencence of the parameters a, b, $SAE_{10,median}$ and $BSF_{min}$ on the number of hourly-averaged samples. The average are the reds circles and the stds the error bars.

**S6. $N_{CCN}$(AOP) calculated by using the site-specific median SAE**

The general combined parameterization was presented in the main test as Eq.10:

$$N_{CCN}(AOP_2) \approx \left( a_1 \ln\left( \frac{SS}{0.093 \pm 0.006} \right)(BSF - BSF_{min}) + R_{min} \right)\sigma_{sp}$$

$$\approx \left( (286 \pm 46)SAE \cdot \ln\left( \frac{SS}{0.093 \pm 0.006} \right)(BSF - BSF_{min}) + (5.2 \pm 3.3) \right)\sigma_{sp}$$

In the main text, we used SAE of hourly-averaged $\sigma_{sp}$ to estimate $N_{CCN}(AOP_2)$. Here we give another alternative for using this formula by using the site-specific median SAE values (Table 4 in the main text). The $N_{CCN}$(AOP) calculated by using the site-specific median SAE is compared with $N_{CCN}$(meas) in Figure SF6. When compared with $N_{CCN}$(AOP) calculated by using the hourly-varying SAE (Fig. 8 in the main text), it is obvious that the two approaches are competitive with each other. A comparison of the biases and correlation coefficients is presented in Table TS3 below. For some combinations of SS and sites, the site-specific median SAE gives a smaller $R^2$ and a higher bias than the hourly SAE especially for ASI.

However, site-specific median SAE is very probably always positive, while the hourly SAE is sometimes negative which may yield negative $N_{CCN}$(AOP). For the 6 sites of this study, the fraction of negative SAE of all hourly data varied between 0-6%. To estimate $N_{CCN}$ for a site with a large fraction of negative SAE, we recommend to use site-specific median SAE.

[Figure]

Figure SF6. Same as Figure 8 in the main text, but $N_{CCN}$(AOP) calculated by using the site-specific

median SAE. For details see the caption of Fig. 8 in the main text

1    Table TS3. Performance of the general combined parametrization using SAE of hourly-
2    averaged scattering coefficients and site-specific median SAE at the supersaturations of the
3    CCN counters of each station.

| Station | Fraction of hourly SAE < 0 | SS | $N_{CCN}(AOP)$ calculated using | | | |
|---|---|---|---|---|---|---|
| | | | hourly-varying SAE | | median SAE | |
| | | | $R^2$ | bias | $R^2$ | bias |
| SMEAR II | 0.0% | 0.10% | 0.675 | 0.72 | 0.657 | 0.72 |
| | | 0.20% | 0.832 | 1.09 | 0.850 | 1.07 |
| | | 0.50% | 0.719 | 1.26 | 0.754 | 1.24 |
| | | 1.00% | 0.504 | 1.20 | 0.554 | 1.18 |
| SORPES | 0.0% | 0.10% | 0.595 | 1.61 | 0.587 | 1.62 |
| | | 0.20% | 0.773 | 1.36 | 0.751 | 1.43 |
| | | 0.40% | 0.650 | 1.22 | 0.699 | 1.30 |
| | | 0.80% | 0.636 | 1.27 | 0.687 | 1.34 |
| MAO | 6.0% | 0.25% | 0.840 | 1.24 | 0.816 | 1.07 |
| | | 0.40% | 0.834 | 0.97 | 0.832 | 0.82 |
| | | 0.60% | 0.725 | 0.91 | 0.742 | 0.76 |
| | | 1.10% | 0.583 | 0.71 | 0.622 | 0.67 |
| PGH | 4.4% | 0.12% | 0.852 | 4.53 | 0.821 | 4.71 |
| | | 0.22% | 0.871 | 2.13 | 0.832 | 2.35 |
| | | 0.48% | 0.784 | 1.13 | 0.779 | 1.30 |
| | | 0.78% | 0.703 | 1.07 | 0.723 | 1.25 |
| ASI | 0.04% | 0.10% | 0.872 | 1.92 | 0.828 | 1.72 |
| | | 0.20% | 0.923 | 1.41 | 0.844 | 1.21 |
| | | 0.40% | 0.900 | 1.61 | 0.836 | 1.35 |
| | | 0.80% | 0.857 | 1.90 | 0.818 | 1.57 |
| PVC | 0.3% | 0.15% | 0.880 | 0.71 | 0.835 | 0.69 |
| | | 0.25% | 0.780 | 0.70 | 0.747 | 0.69 |
| | | 0.40% | 0.687 | 0.71 | 0.655 | 0.69 |
| | | 1.00% | 0.519 | 0.71 | 0.499 | 0.70 |

6    The bias of $N_{CCN}(AOP_2)$ presented in Table TS3 was calculated from the ratio
7    $N_{CCN}(AOP_2)/N_{CCN}(meas)$. Since $N_{CCN}(AOP_2) \approx (a_1 \ln(SS/0.093)(BSF - BSF_{min}) + R_{min})\sigma_{sp}$ it is
8    obvious that biases of $a_1$ affect the bias of $N_{CCN}(AOP_2)$. If we consider the $a_1$ values in the main text
9    Table 4 as the accurate station-specific values then the fitted line $a_1 = 286 \cdot SAE$ overestimates or
10    underestimates $a_1$ by +37%, +30%, -20%, -32%, -20% and +251% for SMEAR II, SORPES,
11    PGH, PVC, MAO and ASI, respectively. These values were calculated from $100\%(286 \cdot SAE -$
12    $a_1)/a_1$. The biases of $a_1$ calculated from $286 \cdot SAE/a_1$ are therefore 1.373, 1.295, 0.796, 0.675,
13    0.792, 3.509 for the respective stations. The average biases of $N_{CCN}(AOP_2)$ at all supersaturations
14    of each station presented in Table TS3 are compared with the biases of $a_1$ in Figure SF7. For each
15    station two values are shown: the average bias of $N_{CCN}(AOP_2)$ calculated by using the median SAE
16    of each station and the average bias of $N_{CCN}(AOP_2)$ calculated by using the hourly-varying SAE.
17    For PGH the average bias of $N_{CCN}(AOP_2)$ at all supersaturations and at SS> 0.3% are shown because
18    the biases at the lowest supersaturations are anomalously high. The plot shows that for most stations
19    the bias of $N_{CCN}(AOP_2)$ can be explained by the bias of $a_1$: when $a_1$ is underestimated so is
20    $N_{CCN}(AOP_2)$ and when $a_1$ is overestimated so is $N_{CCN}(AOP_2)$. PGH is the only exception to this,
21    especially at the lowest two supersaturations (SS = 0.12% and 0.22%) and we cannot explain why.
22    For ASI the bias of $N_{CCN}(AOP_2)$ is clearly smaller than the bias of $a_1$. This would happen when in
23    the formula $N_{CCN}(AOP_2) \approx (a_1 \cdot \ln(SS/0.093)(BSF - BSF_{min}) + R_{min})\sigma_{sp}$ both SAE and BSF are very
24    small and especially when BSF is close to $BSF_{min}$. Both of these would take place when aerosol is
25    dominated by large aerosols. This is true especially for ASI, a site dominated by marine aerosols.

[Figure]

Figure SF 7. Biases of $N_{CCN}(AOP_2)$ vs the bias of $a_1$ calculated from $a_1 = 286 \cdot SAE$. The biases
of $N_{CCN}(AOP_2)$ are the averages of biases at all supersaturations presented in Table TS3. For each
station two values are shown: the average bias of $N_{CCN}(AOP_2)$ calculated by using the median SAE
of each station (open circles) and the hourly-varying SAE (filled circles). For PGH the average bias
of $N_{CCN}(AOP_2)$ at all supersaturations and at SS> 0.3.

---

## Author Response (AR2)

**Revision and replies to the reviewer's comments, 24-10-2019**

We have to thank the reviewer and the editor for their evaluation and their extra voluntary work.

We modified the manuscript to address the reviewer's questions. In the revised text changes are written in red letters. Below the the replies are intended.

**Referee #1:**
**Many thanks for implementing the new Figure 1 - this largely helps to understand the statement on the NCCN-S relationship. However, I still don't agree with the following statement: "The analysis first showed that the dependence of NCCN on supersaturation SS is logarithmic in the range SS < 1.1%.". This infers that finding the ln-relationship is a major discovery. However, there have been several analytical fit functions suggested for the NCCN data over S (Khain et al., 2000; Pinsky et al., 2012; Deng et al., 2013; Pohlker et al., 2016). I wonder why the authors assume that the ln-fit provides advantages over the other functions that have been published before. Moreover, what does "… is logarithmic …" mean? Shouldn't it be rather "… can be described by a logarithmic fit …"? I wonder if the ln-fit has any physical meaning. I suggest that these aspects are discussed in detail in the text. Moreover the aforementioned statement in the abstract probably needs some clarification.**

We agree, the statement of the logarithmic dependence on supersaturation was presented too strongly. There is no theoretical reason for the logarithmic dependence, it is simply a fitting to the data.

The statement
> "The analysis first showed that the dependence of $N_{CCN}$ on supersaturation SS is logarithmic in the range SS < 1.1%. "

in the abstract was replaced according to the reviewer's suggestion by
> "The analysis first showed that the dependence of $N_{CCN}$ on supersaturation SS can be described by a logarithmic fit in the range SS < 1.1%, without any theoretical reasoning.

We thank the reviewer for the new reference list. It revealed that we had not sufficiently read the existing literature. We now read the papers and analyzed the parameterizations presented in them and added related discussion in sections 3.1 and 3.2.

In section 3.1 we added the text below in red letters:

> ### 3.1 Site-dependent $N_{CCN}$ - AOP relationships
> The averages of AOPs of $PM_{10}$ particles and $N_{CCN}$ at four supersaturations during the analyzed period for each site are presented in Table 2. In general all of them are cleaner than SORPES and more polluted than SMEAR II, based on the average values of $\sigma_{sp}$. The average values of $N_{CCN}$ are obviously higher in more polluted air as well as can be seen in the values presented in Table 2. The dependence of $N_{CCN}$ on SS is shown by plotting the averages of the measured $N_{CCN}$ at the six sites at the station-specific supersaturations of the CCN counters (Fig. 1). In all these different types of environments a logarithmic function fits better to the data than the power function

$N_{CCN}(SS) = C \times (SS)^k$. It is not a new observation that the power function is not perfect for describing the $N_{CCN}$ vs. SS relationship. Also other function types have been used in the literature, for instance a product of the power function and the hypergeometric function (Cohard et al., 1998; Pinsky et al., 2012), an exponential function (Ji and Shaw, 1998; Mircea et al., 2005; Deng et al., 2013) and the error function (e.g., Dusek et al., 2003 and 2006b; Pöhlker et al., 2016). In the following analysis of the relationships between $N_{CCN}$, AOPs and SS we will use logarithmic fittings to the data without any theoretical reasoning.

In section 3.2 we added the text below in red letters. There is an argumentation for using the logarithmic fittings even though there is no theoretical explanation.

The above derivation of the combined parameterization by using the logarithms of SS was fairly straightforward. In the error-function parameterizations of Dusek et al. (2003) and Pöhlker et al. (2016) there are adjustable parameters that affect the argument of the error function. In the parameterization of Ji and Shaw (1998) there is an exponential function where the argument contains the power function of SS and the parameterization of by Cohard et al. (1998) is a product of the power function and the hypergeometric function. If these functions were used for fitting the $N_{CCN}$(AOP, SS) data it would be would be more complicated to combine the site-dependent parameterizations into a general equation analogous to Eq. (8). The simplicity of the logarithmic fitting makes it most suitable for our approach. The disadvantage of Eq. (8) is that it predicts no upper limit for $N_{CCN}$ at high supersaturations. This is not correct since $N_{CCN}$ cannot be larger than the total particle number concentration and therefore it has to be emphasized that the parameterization presented here is only valid in the range of SS < 1.1%.

In the conclusions we wrote

A logarithmic function was fitted to the $N_{CCN}$ vs. supersaturation SS data in the range SS < 1.1%. For $N_{CCN}$(AOP) the fitting yielded a logarithmic dependence on SS: $N_{CCN}$(AOP) $\approx$ $(286 \cdot SAE \cdot \ln(SS/0.093)(BSF - BSF_{min}) + (5.2 \pm 3.3))\sigma_{sp}$.